# Conserved shifts in sperm small non-coding RNA profiles during mouse and human aging

Junchao Shi [1,2,11,13], Xudong Zhang [1,2,13], Chen Cai[1], Shichao Liu [1], Jiancheng Yu[3], Emma R James[1], Lihua Liu[1], Benjamin R Emery[1], Megan R McMurray Bires [1], Elizabeth Torres-Arce[1,12], Hukam C Rawal[4], Joemy Ramsay[1], Jason Kunisaki[3], Changcheng Zhou [2], David S Milstone [5], Mary Elizabeth Patti[6], Xiaoxu Yang [3], Tim G Jenkins[1,7], Aaron Quinlan[3], Bradley R Cairns[8], Paul Schimmel[9], James M Hotaling[1,10], Kenneth I Aston [1✉], Tong Zhou [4✉] & Qi Chen [1,2,3✉]

## Abstract

Sperm aging impacts male fertility and offspring health, highlighting the need for reliable aging biomarkers to guide reproductive decisions. However, the molecular determinants of sperm fitness during aging remain ill-defined. Here, we profiled sperm small non-coding RNAs (sncRNAs) using PANDORA-seq, which overcomes RNA modification–induced detection bias to capture previously undetectable sncRNA species associated with mouse and human spermatozoa throughout the lifespan. We identified an "aging cliff" in mouse sperm RNA profiles—a sharp age-specific transition marked by significant shifts in genomic and mitochondrial tRNA-derived small RNAs (tsRNAs) and rRNA-derived small RNAs (rsRNAs). Notably, rsRNAs in mouse sperm heads exhibited a transformative length shift, with longer rsRNAs increasing and shorter ones decreasing with age, suggesting altered biogenesis or processing with age. Remarkably, this sperm head-specific shift in rsRNA length was consistently observed in two independent human aging cohorts. Moreover, transfecting a combination of tsRNAs and rsRNAs resembling the RNA species in aged sperm was able to induce transcriptomic changes in mouse embryonic stem cells, impacting metabolism and neurodegeneration pathways, mirroring the phenotypes observed in offspring fathered by aged sperm. These findings provide novel insights into longitudinal dynamics of sncRNAs during sperm aging, highlighting an rsRNA length shift conserved in mice and humans.

Keywords Sperm Epigenetics; Aging Clock; Sperm RNA Code; Paternal Age Effect; Epigenetic Inheritance
Subject Categories Methods & Resources; RNA Biology; Stem Cells & Regenerative Medicine

## Introduction

In humans, fathers of advanced age are on the rise (Khandwala et al, 2017). Advanced paternal age not only compromises male fertility (Jimbo et al, 2022) but also poses risks to offspring health, associated with increased risks of stillbirth and a range of complications in subsequent generations, including elevated susceptibility to developmental, neuropsychiatric, and behavioral anomalies (Khandwala et al, 2018; Taylor et al, 2019). Beyond epidemiological evidence, rodent models have further reinforced the connection between paternal age and elevated risks of metabolic disorders, obesity, and anxiety-related behaviors (Guo et al, 2021; Mao et al, 2022). Traditionally, sperm aging research has focused on DNA integrity and methylation patterns (Ashapkin et al, 2023; Jenkins et al, 2014; Kunisaki et al, 2024; Neville et al, 2025; Seplyarskiy et al, 2025; Watson, 2025). However, recent discoveries have increasingly highlighted the epigenetic potential of mammalian sperm small non-coding RNAs (sncRNAs), including miRNAs, tRNA-derived small RNAs (tsRNAs), rRNA-derived small RNAs (rsRNAs), and their associated RNA modifications in mediating the intergenerational transmission of paternal environmental clues to offspring (Chen et al, 2016a; Chen et al, 2016b; Donkin et al, 2016; Gapp et al, 2014; Liu et al, 2023; Natt et al, 2019; Sarker et al, 2019; Sharma et al, 2016; Zhang et al, 2021; Zhang et al, 2018), including mediating age-related traits to the offspring (Guo et al, 2021; Liang et al, 2022; Miyahara et al, 2023). Sperm sncRNAs can also serve as biomarkers for embryo quality in IVF clinics (Hua et al, 2019;

[1]Molecular Medicine Program, Division of Urology, Department of Surgery, University of Utah School of Medicine, Salt Lake City, UT, USA. [2]Division of Biomedical Sciences, Center for RNA Biology and Medicine, School of Medicine, University of California, Riverside, CA, USA. [3]Department of Human Genetics, University of Utah School of Medicine, Salt Lake City, UT, USA. [4]Department of Physiology and Cell Biology, University of Nevada, Reno School of Medicine, Reno, NV, USA. [5]Department of Pathology, Brigham and Women's Hospital and Harvard Medical School, Boston, MA, USA. [6]Joslin Diabetes Center, Harvard Medical School, Boston, MA, USA. [7]Department of Cell Biology and Physiology, Brigham Young University, Provo, UT, USA. [8]Howard Hughes Medical Institute, Department of Oncological Sciences and Huntsman Cancer Institute, University of Utah School of Medicine, Salt Lake City, UT, USA. [9]Department of Molecular Medicine, The Scripps Research Institute, La Jolla, CA, USA. [10]Induction Bio, Salt Lake City, UT, USA. [11]Present address: China National Center for Bioinformation and Beijing Institute of Genomics, Chinese Academy of Sciences, Beijing, China. [12]Present address: School of Environmental and Life Sciences, The University of Newcastle, Callaghan, and Infertility and Reproductive Research Program, Hunter Medical Research Institute, New Lambton Heights, NSW, Australia. [13]These authors contributed equally: Junchao Shi, Xudong Zhang. ✉E-mail: ki.aston@hsc.utah.edu; tongz@med.unr.edu; qi.chen@hsc.utah.edu

Isacson et al, 2025). This has given rise to the concept of 'sperm RNA code' (Zhang et al, 2019), proposing that specific sperm RNA signatures, reflecting paternal experiences, play a critical role in controlling offspring health.

To decode the sperm RNA code, we developed PANDORA-seq (Shi et al, 2021), a novel method that overcomes limitations in traditional sncRNA sequencing by addressing RNA modifications (Shi et al, 2022), enabling comprehensive analysis of previously undetectable sncRNAs, particularly tsRNAs and rsRNAs with such modifications (Shi et al, 2021). PANDORA-seq has revolutionized our understanding of the sncRNA landscape in sperm, uncovering that miRNAs account for less than 1% of sperm sncRNAs, while tsRNAs and rsRNAs emerge as the dominant sperm sncRNAs that play critical roles in sperm-mediated epigenetic inheritance (Chen, 2022).

In this study, PANDORA-seq revealed a previously undetected 'aging cliff' in mouse sperm aging—a sharp transition in tsRNA and rsRNA profiles that traditional sncRNA-seq could not detect. Furthermore, analysis of sncRNAs in sperm heads uncovered an age-dependent length shift in rsRNAs, a phenomenon observed in both mice and humans. These findings suggest conserved mechanisms of sncRNA processing during aging, holding the potential for developing clinical biomarkers to assess human sperm aging and quality.

## Results

### PANDORA-seq reveals a sharp tsRNA/rsRNA "aging cliff" in mouse sperm

To analyze the sperm aging process in mice, we utilized the inbred strain C57BL/6J from Jackson Laboratory, organized into five age groups at 20-week intervals (10-, 30-, 50-, 70-, and 90-week old) (Fig. 1A). Mature sperm from the cauda epididymis (four mice per age group) were isolated for RNA extraction as previously described (Peng et al, 2012). The extracted sperm RNA from each individual was split into two portions: one for traditional sncRNA-seq and the other for PANDORA-seq. Sequencing data were annotated for different sncRNA types (e.g., miRNAs, tsRNAs, and rsRNAs) using the SPORTS tool (Shi et al, 2018) and further analyzed for age-related changes in sncRNA profile (see Methods). Through principal coordinate analysis (PCoA) of PANDORA-seq data (Fig. 1B), we observed a distinct 'aging cliff' between 50- and 70-week intervals, marked by a dramatic shift in tsRNA/rsRNA composition (computed across all tsRNA and rsRNA categories). This stark change delineated early (10–50 weeks) from late (70–90 weeks) stages, unlike traditional sncRNA-seq (Fig. 1B), which showed subtler changes, likely due to missed detection of highly modified sncRNAs.

In addition, we conducted a similar PCoA based on sperm miRNA composition by computing every miRNA expression. Intriguingly, even though miRNA reads constitute <0.5% of total reads in PANDORA-seq compared to >5% in traditional sncRNA-seq (Table EV1), the miRNA profile from PANDORA-seq still delineated an aging cliff between the 50- and 70-week intervals, which was not observed in traditional sncRNA-seq (Fig. EV1A–C). One possible explanation for this observation is that a subset of the miRNAs detected by PANDORA-seq are actually derived from

tsRNAs and rsRNAs, a phenomenon we have previously reported (Shi et al, 2021). This result is somewhat unexpected given the low proportion of miRNA reads in the PANDORA-seq dataset, but it highlights the superior sensitivity of PANDORA-seq in detecting age-related changes within sncRNA populations, even when these changes are represented by a relatively minor fraction of the total reads. However, it should be noted that the 'aging cliff' identified by miRNAs between the 50- and 70-week intervals is less pronounced than that revealed by tsRNAs/rsRNAs. This difference is supported by the ratio of between-group variance to within-group variance (*F*-statistic) (Fig. EV1D), which is significantly higher for tsRNAs/rsRNAs compared with miRNAs, supporting a better separation between the demarcated stages (early 10-/30-/50-week vs. later 70-/90-week) using tsRNA/rsRNA profiles.

Since the primary part of sperm for fertilization is the information stored in sperm head, containing both the DNA and RNAs that are deeply embedded in the nuclei, these head-embedded RNAs are potentially more functionally relevant during the early epigenetic reprogramming events in the male pronucleus and in regulating embryo development (Chen, 2022). By contrast, RNAs present in the limited sperm cytoplasm are rapidly diluted in the oocytes at fertilization. Indeed, our previous study showed that the de-membranated sperm heads contain different sncRNA profiles compared to the intact sperm (Shi et al, 2021). Given the sperm head's unique sncRNA profile and function, we further performed PANDORA-seq and traditional sncRNA-seq on de-membranated mouse sperm heads across the same age groups as studied with whole sperm (10-, 30-, 50-, 70-, and 90-week-old). While there are differences in the sncRNA composition between sperm heads and whole sperm, both sample types revealed aging signatures indicative of an "aging cliff" at the 50–70 week transition for both tsRNA/rsRNA (Fig. 1C) and miRNAs (Fig. EV1E–H) using PANDORA-seq, a pattern not clearly observed with traditional sncRNA-seq. Individual tsRNA/rsRNA mappings for intact sperm (Fig. 1D) and heads (Fig. 1E) show substantial 50–70 week alterations, recapitulating the aggregate cliff by overall tsRNA/rsRNA profile.

### Mitochondrial tsRNAs/rsRNAs in sperm heads parallel genomic sncRNA aging patterns

Interestingly, although the sperm head samples are completely depleted of sperm tail (Fig. 1F) and thus no mitochondria are included, we consistently detected mitochondrial tsRNAs (mt-tsRNAs) and mitochondrial rsRNAs (mt-rsRNAs) in the sperm heads. This resonates with the recent reports that mt-tsRNAs/mt-rsRNAs are sensitive to dietary stress (Cai and Chen, 2024; Natt et al, 2019; Ramesh et al, 2023) and act as signal molecules in controlling embryo development (Cai and Chen, 2024). We found that while the percentages of mitochondrial tsRNAs and rsRNAs are very low in the sperm head (0.14 and 0.11% respectively) (Fig. 1F), their levels are highly correlated, with coordinated up- or downregulation across age groups (Figs. 1G and EV2). Moreover, despite the low expression level of mitochondrial tsRNAs and rsRNAs compared to the genomic tsRNAs and rsRNAs, they effectively distinguish age groups, mirroring the genomic tsRNA/rsRNA aging cliff (Fig. 1H). This finding suggests that mitochondrial tsRNAs and rsRNAs, despite their low abundance, encode aging-relevant information that parallels genomic tsRNAs/rsRNAs.

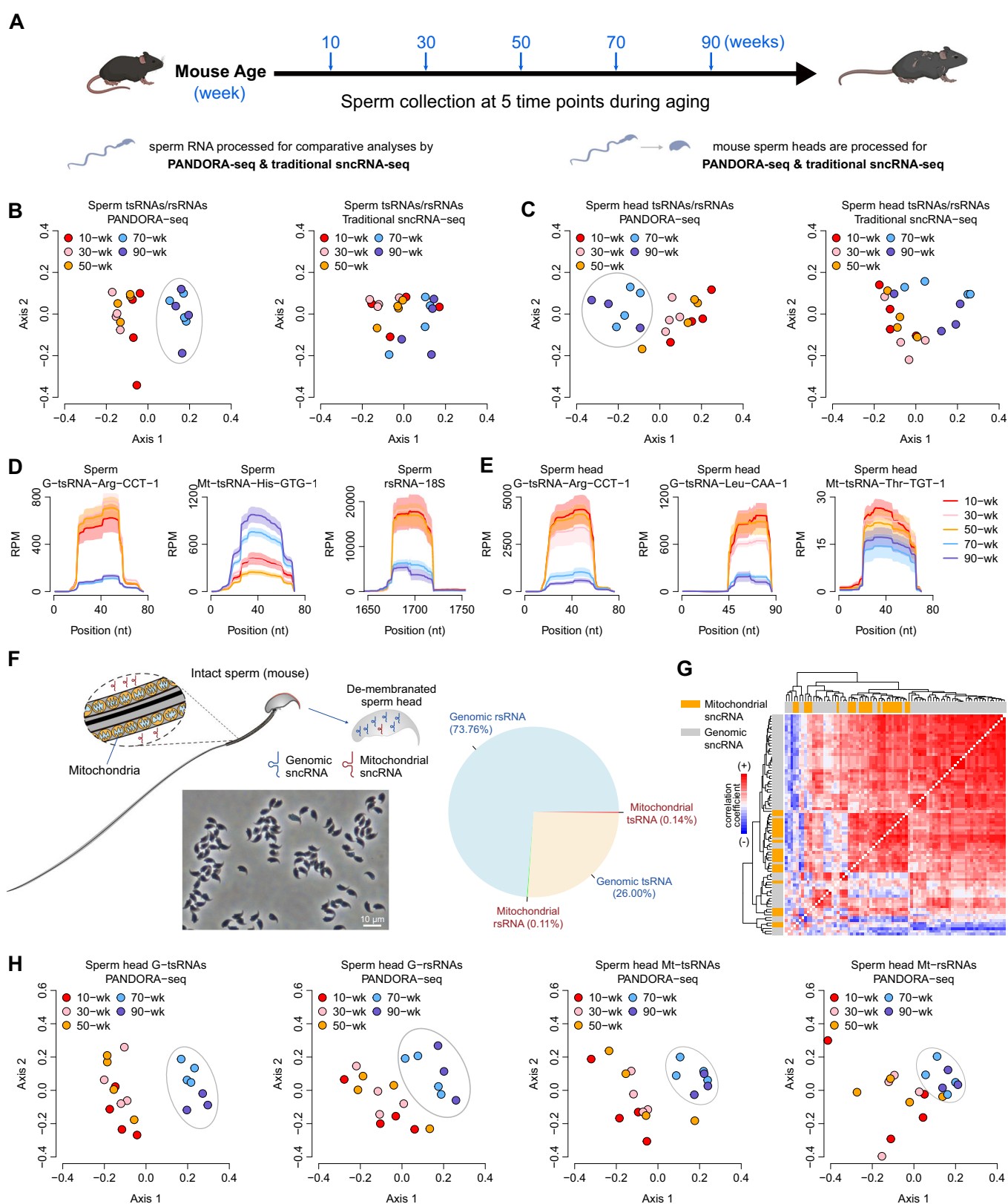

**Figure 1.   Discovery of an 'aging cliff' in mouse sperm sncRNA profiles using PANDORA-seq.**

(A) Schematic of the experimental design: Sperm collected from C57BL/6J mice at 5 time points (10-, 30-, 50-, 70-, and 90-week; $n = 4$ per group) during aging, processed for comparative analyses by PANDORA-seq and traditional sncRNA-seq on intact sperm and de-membranated sperm heads. (B) Principal coordinate analysis (PCoA) of mouse intact sperm tsRNA/rsRNA data showing that PANDORA-seq, but not traditional sncRNA-seq, identified a clear 'aging cliff' separation during the 50–70-week transition in mouse sperm, marked by a dramatic shift in tsRNA/rsRNA composition. Axis 1: the first principal coordinate; Axis 2: the second principal coordinate. (C) PCoA on sperm head tsRNA/rsRNA data similarly show clearer demarcation during the 50–70-week transition using PANDORA-seq, compared to traditional sncRNA-seq. The PCoA presented in panel (B, C) was performed based on the expression profile of the individual tsRNA/rsRNA families. (D, E) Coverage plots of illustrative tsRNAs and rsRNAs from (D) intact sperm and (E) sperm heads mapped to their parent sequences across age groups. The reads mapping also shows the location from which the tsRNAs and rsRNAs are derived from the mature tRNAs and rRNAs. The solid curves indicate the mean of reads per million (RPM), while the shaded bands indicate the standard error of the mean. (F) Schematic illustrating de-membranation of sperm and the image of purified sperm heads (depleted of tails and mitochondria), with pie chart showing the composition of sncRNAs in sperm heads: predominantly genomic rsRNAs (73.7%) and tsRNAs (26.0%), with minor mitochondrial tsRNAs (0.14%) and rsRNAs (0.11%). (G) Correlation heatmap of all genomic tsRNA/rsRNA and mitochondrial tsRNA/rsRNA in the de-membranated sperm heads. The colors in the heatmap represent the intensity of co-expression (i.e., Spearman's rank correlation coefficient) between the sncRNAs: red indicates positive co-expression while blue indicates negative co-expression. Notably, the highly positively correlated clusters in the center square area (*Fisher*'s exact test: $P = 9.4 \times 10^{-8}$) are largely overlapped with mt-tsRNAs/rsRNAs, marked in orange color. The detailed identity of each tsRNAs and rsRNA category is shown in Fig. EV2. (H) Separated PCoAs for genomic tsRNAs, genomic rsRNAs, mitochondrial tsRNAs, and mitochondrial rsRNAs in sperm heads across the aging process. Mitochondrial tsRNAs and rsRNAs show demarcation power similar to genomic tsRNAs/rsRNAs, despite their relatively low expression level. The PCoA was performed based on the expression profile of the individual sncRNA species. G-tsRNA (genomic tsRNA), Mt-tsRNA (mitochondrial tsRNA), G-rsRNA (genomic rsRNA), Mt-rsRNA (mitochondrial rsRNA). Source data are available online for this figure.

The consistent detection of mt-tsRNAs and mt-rsRNAs in de-membranated sperm heads also strongly suggests that these RNA fragments are transported from the mitochondria to the nucleus, potentially serving as a mechanism for mitochondria-nucleus communication that influences the sperm aging process, a possibility that warrants further investigation.

## Sperm-head rsRNAs exhibit an age-dependent length shift in mice

Beyond the general aging cliff, PANDORA-seq analysis of sncRNA subtypes in mouse sperm revealed an unexpected age-related shift: the relative abundance of longer rsRNAs increased while that of shorter rsRNAs decreased (Fig. 2A,C)—based on the calculation of the association of expression (*RPM*) of each RNA length category with age, which we termed as aging index ($I_a$) (Fig. 2A). This length shift in rsRNAs is specific to sperm heads (Fig. 2A), not whole sperm (Fig. 2B), and is prominent in rsRNAs derived from 28S- and 18S-rRNAs (Fig. 2A), which dominate the rsRNA pool. Similar trends are also observed in mitochondrial rsRNAs derived from 12S-rRNA and 16S-rRNA (Fig. EV3A). In contrast, tsRNAs did not show such an age- associated length-shift pattern overall (Fig. EV4).

The rsRNA length shift suggests that aged sperm have a reduced ability to process longer rsRNAs into shorter fragments, possibly due to oxidative-stress-induced changes that alter the activity or abundance of the responsible enzymes during aging. Supporting this, we identified regions where longer rsRNAs accumulate, and shorter rsRNAs diminish during aging, such as in the 4660–4730 nt of 28S-rRNA (Fig. 2D). Notably, these longer 28S-rsRNAs are among the most abundant sncRNAs showing significant age-related changes. This finding raises the possibility that longer and shorter rsRNAs play distinct roles, potentially contributing to reduced fertility or offspring health issues linked to aged sperm.

## A conserved rsRNA length shift marks human sperm aging

Given PANDORA-seq's success in detecting aging signatures in mouse sperm, we next applied it to human sperm from two

independent cohorts: a longitudinal cohort of 8 donors, each providing two samples collected over 6–23 years apart, with ages ranging from 34 to 68 years (cohort-1) (Fig. 3A); and a cross-sectional cohort of 47 donors aged 25–51 years (cohort-2) (Fig. 3B). For both cohorts, we isolated RNA from de-membranated sperm heads for two reasons: (1) mouse data showed that sperm heads are enriched with aging-related signals, including the rsRNA length shift, and (2) human sperm variability—such as viscosity, debris, or cytoplasmic droplets—can confound RNA purity, making sperm heads a more reliable source of uncontaminated RNA.

Analysis of PANDORA-seq data from both human cohorts revealed a consistent age-related shift in rsRNA length, mirroring the trends observed in mice (e.g., longer rsRNAs increasing and shorter decreasing in relative abundance). This shift was evident in total rsRNAs (Fig. 3A,B), and specifically for rsRNAs derived from 18S- and 28S-rRNAs (Fig. 3A,B). Mitochondrial rsRNAs also showed a similar trend but with weaker statistical significance (Fig. EV3B,C), likely due to their lower abundance and fewer sequencing reads, compounded by the inherent genetic and environmental variability of human samples. Nonetheless, these findings suggest that the rsRNA length shift is an evolutionarily conserved feature of aging across mice and humans. This shift may stem from age-related changes in enzymatic activity or levels, potentially linked to altered mitochondrial function and oxidative stress—common factors in sperm aging (Aitken, 2023). Importantly, it is well established that oxidative stress controls the fragmentation of tRNA and rRNA by regulating the recruitment and activity of specific ribonucleases (Chen et al, 2021). Such alterations in these processes impact the biogenesis and/or processing of tsRNAs/rsRNAs and reshape the "sperm RNA code," which may impact embryo development and offspring phenotypes as a functional readout of sperm aging.

## Age-mimicking sncRNA profiles reprogram embryonic metabolic and neurodegenerative pathways

To investigate the functional significance of age-related changes in sperm sncRNAs, we selected a group of tsRNAs and rsRNAs that exhibited high expression level with the most significant differences in expression shared in sperm and sperm heads during mouse aging

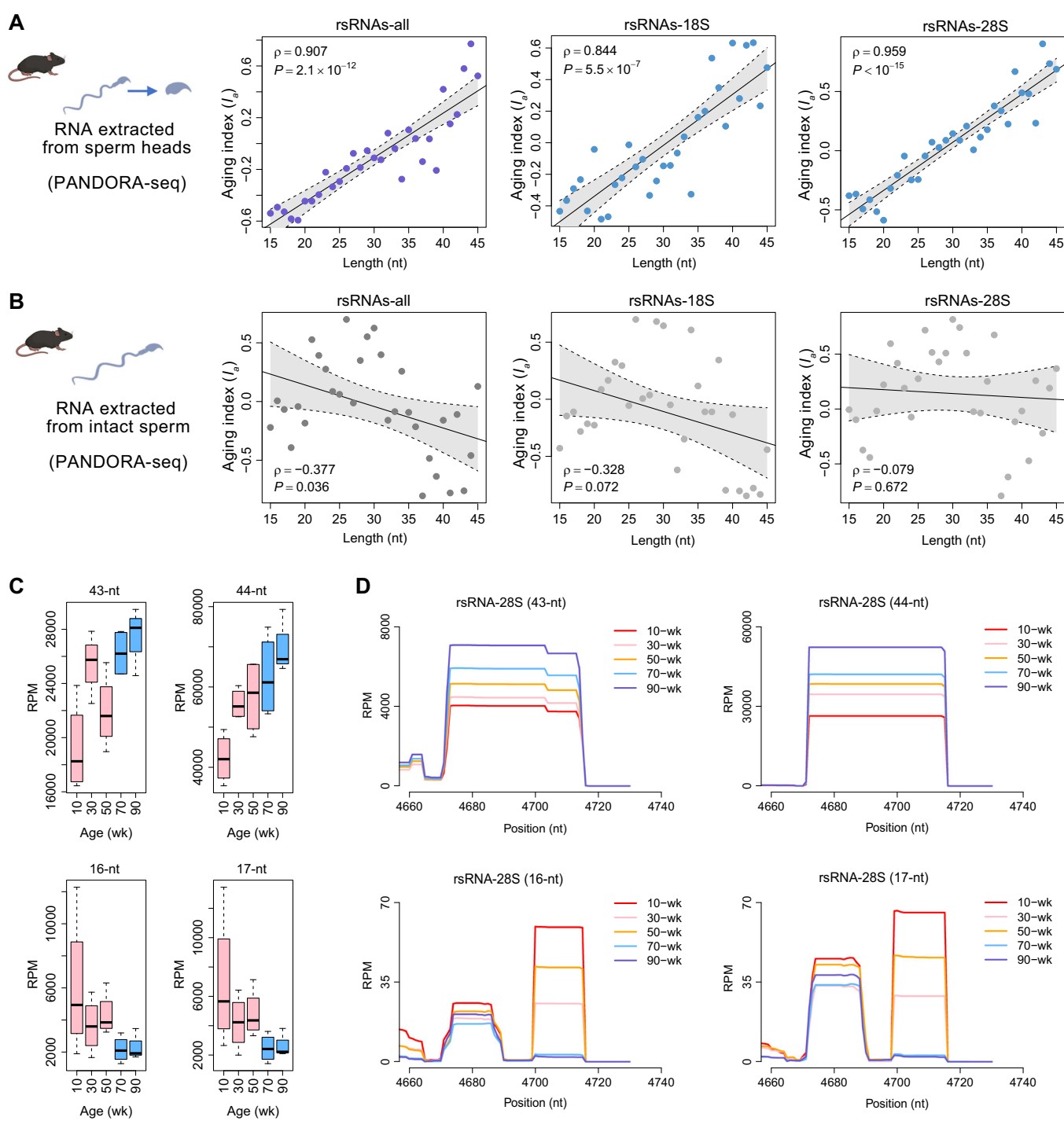

(Fig. 4A). These RNAs were combined into cocktails representing "young" and "old" sperm profiles (see Fig. 4A and Methods for detailed composition). We transfected the RNA cocktails into mouse embryonic stem cells (mESCs), chosen as a model system because they resemble early embryonic cells potentially influenced by sperm-derived sncRNAs during pre-implantation development. To assess the functional impact of these age-related sncRNA profiles, we performed RNA-seq analysis at 24 h post-transfection in mESCs, a time point at which no apparent growth or morphological changes were observed.

As a result, principal component analysis (PCA) revealed clear separation of gene expression profiles among the vehicle (mock-transfected), young-combo, and old-combo groups (Fig. 4B). Both RNA-treated groups differed substantially from the vehicle, with overlap in altered genes as shown in Venn diagram (Fig. 4C; Tables EV2–5) yet also showed distinct gene expression profiles between

**Figure 2.  Discovery of age-related length shift of rsRNAs in mouse sperm heads.**

(A) Age-related length shift of total rsRNAs, 18S-derived rsRNAs, and 28S-derived rsRNAs in mouse sperm head. For each RNA length category, we calculated the association of expression (*RPM*) with age (*Spearman*'s rank correlation), which we termed as aging index ($I_a$). Each dot represents the value of $I_a$ for the corresponding length. The scatter plots demonstrate the relationship between $I_a$ and RNA length, which was measured by *Spearman*'s rank correlation coefficient ($\rho$) and the corresponding *P* value. The solid lines depict linear regression fits. Strong positive correlations were observed between $I_a$ and RNA length across all the categories (i.e., total rsRNA and 18S-/28S-derived rsRNAs). (B) Relationship between $I_a$ and RNA length in mouse intact sperm. Weak negative correlation (in total rsRNA) or no correlations (in 18S-derived and 28S-derived rsRNAs) were observed regarding length shifts with age. (*Spearman's* rank correlation). (C) Boxplot of *RPM* for different rsRNA lengths (43-nt, 44-nt, 16-nt, and 17-nt) from total rsRNAs across mouse age groups (10–90 weeks). The bold horizontal line indicates the median of the data. The lower and upper boundaries of the box indicate the 25th percentile (Q1) and 75th percentile (Q3) of the data, respectively. Accordingly, the interquartile range (IQR) is Q3-Q1. The lower and upper whiskers indicate the most extreme data points that fall within Q1-1.5 × IQR and Q3 + 1.5×IQR, respectively. (D) Coverage plots of the rsRNA fragments mapped to 28S-rRNA (4660 nt - 4730 nt), stratified by length (43-nt, 44-nt, 16-nt, and 17-nt). Lines represent the mean coverage level at each position, illustrating the accumulation of longer fragments and diminution of shorter ones with advancing age. Source data are available online for this figure.

the old-combo and young-combo, as shown in the heatmap (Fig. 4D; Tables EV6 and 7).

Further Gene Ontology analyses of the differentially expressed genes between young-combo and old-combo groups revealed that the old-combo induced upregulation of genes involved in metabolic pathways (e.g., fatty acid metabolism, carbon metabolism, glycolysis/gluconeogenesis), mitochondrial function (oxidative phosphorylation, mitophagy), and neurodegenerative diseases (e.g., Parkinson's disease, Alzheimer's disease, Huntington's disease) (Fig. 4E; Table EV8). These altered pathways align with offspring phenotypes associated with aged sperm or induced by zygotic injection of RNAs from aged sperm, including metabolic and neurological disorders (Guo et al, 2021; Mao et al, 2022). Interestingly, recent reports have shown that dysregulated sperm RNAs under high-fat diet conditions can impact the metabolic gene expression in the early embryo (Chen et al, 2016a; Tomar et al, 2024), especially impacting two-cell embryo's oxidative phosphorylation pathway (Cai and Chen, 2024; Tomar et al, 2024), suggesting an intertwined relationship between aging and dietary factors on sperm quality and the subsequent mitochondrial functions in embryo development and offspring health. Our findings here provide proof-of-principle evidence that different combinations of tsRNAs/rsRNAs mimicking young versus old sperm status can profoundly affect mESC gene expression, the mechanistic details of which await further investigations, and the RNA modification status of tsRNAs/rsRNAs may further complicate the situation (Chen and Zhou, 2023; Kuhle et al, 2023; Shi et al, 2022).

## Discussion

Our discovery of the sncRNA-based "aging cliff" in mouse sperm characterizes a striking shift in tsRNA and rsRNA profiles within a specific age range. However, what drives this sudden transition? Studies of other systems, like blood proteins, reveal similar cliff-like aging patterns (Ding et al, 2025; Shen et al, 2024), reinforcing the idea that aging can leap forward at critical time windows. Does this suggest that molecular changes occur abruptly at a certain stage? Or could subtle, progressive changes act as "forerunner signals" that set the stage for the later, larger shift? In our data, despite the overall tsRNA/rsRNA aging cliff, some sperm tsRNA/rsRNA levels indeed change gradually; in particular, the rsRNA length shift progresses steadily during mouse aging (Fig. 2D). Our mESC transfection experiments further show that

sncRNAs exhibiting these gradual changes, when introduced, trigger substantial transcriptomic responses, particularly in mitochondrial function pathways (Fig. 4E). These pathways, central to cellular aging, might amplify subtle sncRNA shifts into the pronounced cliff we observe, suggesting a model where incremental changes (Shi et al, 2015) (e.g., the linear trends of quantity and length shift of individual rsRNAs) potentially driven by accumulating oxidative stress, cascade into a transformative cliff effect.

The source of these sperm sncRNA changes remains an open question. They may arise from altered biogenesis and regulation during testicular germ cells development, epididymal sperm maturation, or from sncRNAs delivered by somatic cells reflecting systemic aging, or from a combination of these factors (Chen et al, 2016b; Conine and Rando, 2022; Cui et al, 2025; Nie et al, 2022). While the precise upstream trigger of aging is unclear, gradual oxidative stress—a hallmark of aging in the testis, sperm, and other systems—likely plays a significant role. Oxidative stress is known to modulate tRNA and rRNA fragmentation by regulating ribonuclease activity at specific positions (Chen et al, 2021; Thompson et al, 2008), potentially driving the observed shifts in tsRNA and rsRNA level and length, the details of which warrant further investigations.

Finally, the mechanisms by which sperm tsRNAs and rsRNAs regulate gene expression in mESCs and/or early embryonic development remain underexplored. While certain tsRNAs and rsRNAs can use miRNA-like mechanisms to linearly bind to their RNA targets, with or without the involvement of AGO proteins (Chen and Zhou, 2023), other tsRNAs and rsRNAs may fold into specific structures, to bind to specific proteins to exert their functions in an aptamer-like manner (Chen and Zhou, 2023; Kuhle et al, 2023). These modes of action can be further fine-tuned by specific RNA modifications, which can regulate sncRNA stability (Zhang et al, 2018), linear binding efficiency to their RNA targets (Su et al, 2022) or protein partners (Guzzi et al, 2018), enabling increased functional versatility. Due to the complex set of RNA modifications carried by tsRNAs and rsRNAs, one limitation of our transfection experiment (Fig. 4A) is that these synthetic tsRNAs and rsRNAs may not function in a way that fully mimics their in vivo status. Future methods like nanopore-based sequencing (Lucas et al, 2023) and advanced mass spectrometry-based methods (Yuan et al, 2024), after proper optimization (Shi et al, 2022), may provide detailed RNA modification maps for each sncRNA, and thus guide more precise functional investigations.

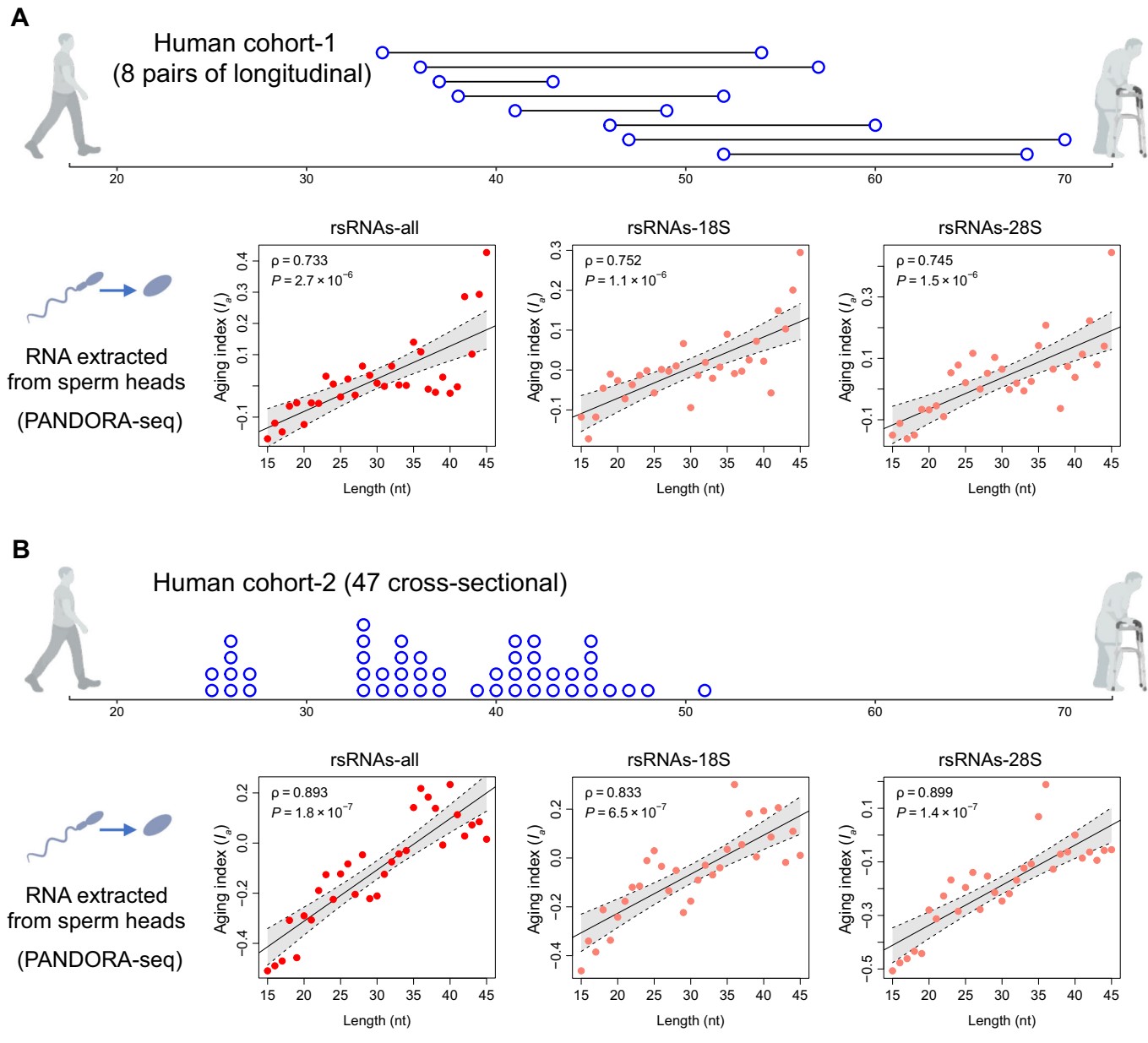

**Figure 3. Conserved age-related length shift of rsRNAs in human sperm cohorts.**

(A) Age-related length shift of total rsRNAs, 18S-derived rsRNAs, and 28S-derived rsRNAs in human sperm head from cohort-1. Upper panel shows the schematic of the longitudinal human cohort-1 (eight donors, each with paired samples over 6–23 years apart, with ages ranging from 34 to 68 years. The blue hollow dots connected by line represent each individual donor. RNA sample was extracted from de-membranated sperm heads followed by PANDORA-seq. For each RNA length category, we calculated the association of expression (RPM) with age (Spearman's rank correlation), which we termed as aging index ($I_a$). Each dot represents the value of $I_a$ for the corresponding length. The scatter plots demonstrate the relationship between $I_a$ and RNA length, which was measured by Spearman's rank correlation coefficient ($\rho$) and the corresponding P value. The solid lines depict linear regression fits. (B) Age-related length shift of total rsRNAs, 18S-derived rsRNAs, and 28S-derived rsRNAs in human sperm head from the cross-sectional cohort-2 (47 donors with ages ranging from 25 to 51 years). The blue hollow dots represent individual donors. The scatter plots demonstrate the consistent relationship between $I_a$ and RNA length in cohort-2 .Both human cohorts show conserved age-related length shift of rsRNAs (total rsRNAs, rsRNA-18S, and rsRNA-28S), which mirrors the observation in mouse aging. (Spearman's rank correlation). Source data are available online for this figure.

In summary, leveraging the advances of PANDORA-seq, we have unveiled a novel sperm aging landscape in mice and humans. The conserved rsRNA length shift across species signals the potential for sncRNA-based biomarkers to guide informed reproductive decisions. Looking ahead, tracing the origins of these sncRNA changes and identifying the RNA-processing enzymes that control tsRNA and rsRNA biogenesis and length could unlock transformative interventions. These directions may not only predict but also prevent the transmission of age-related disorders to offspring, heralding a new era in reproductive health where a deeper understanding of the "sperm RNA code" helps shape healthier generations.

# Methods

### Reagents and tools table

| Reagent/resource | Reference or source | Identifier or catalog number |
|---|---|---|
| **Experimental models** | | |
| C57BL/6J mice | The Jackson Laboratory | IMSR_JAX:000664 |
| Mouse embryonic stem cell (v6.5) | Dr. Sihem Cheloufi Lab | |
| **Recombinant DNA** | | |
| **Antibodies** | | |
| **Oligonucleotides and other sequence-based reagents** | | |
| tsRNA-Pro-CGG: 5'/Phos/ rGrGrCrUrCrGrUrUrGr GrUrCrUrArGrGrGrGrUrArUr GrArUrUrCrUrCrGrCrUrU-2'3 c-Phos | ChemGene | Customer service, Lot#: L193598-1 |
| 28S-rsRNA 44-nt: 5'/Phos/ rCrUrCrGrCrUrGrCr GrArUrCrUrArUrUrGrArArArGr UrCrArGrCrCrCrUrCrGrArCrArCrAr ArGrGrGrUrU rUrG-2'3 c-Phos | ChemGene | Customer service, Lot#: L193598-2 |
| 28S-rsRNA 17-nt: 5'/Phos/ rCrUrCrGrArCrArCrArArAr GrGrGrUrUrG-2'3 c-Phos | ChemGene | Customer service, Lot#: L193598-3 |
| tsRNA-Arg-CCT: 5'/Phos/ rArGrGrArUrUrGrUrGrGr GrUrUrCrGrArGrUrCrCrCr ArUrCrUrGrGrGrUrGrC-2'3 c-Phos | ChemGene | Customer service, Lot#: L193598-4 |
| 5S-rsRNA: 5'/Phos/rUrGr GrArGrArCrCrGrCrCrUrGr GrGrArArUrArCrCrGrGrGr UrGrCrUrGrUrArGrGrCrU-2'3 c-Phos | ChemGene | Customer service, Lot#: L193598-5 |
| **Chemicals, Enzymes and other reagents** | | |
| AlkB enzyme | Addgene | 228218 |
| T4PNK | New England Biolabs | M0201L |
| Lipofectamine Stem Transfection Reagent | Invitrogen | STEM00003 |
| NEBNext® Small RNA Library Prep Set for Illumina® (Multiplex Compatible) | New England Biolabs | E7330S |
| TRIzol | Invitrogen | 15596018 |
| 10 X TBE | Invitrogen | AM9863 |
| 2X RNA loading dye | New England Biolabs | B0363S |
| Urea | Invitrogen | AM9902 |
| Ammonium persulfate | Sigma-Aldrich | A3678-25G |
| TEMED | Thermo Fisher Scientific | BP150-100 |
| SYBR Gold | Invitrogen | S11494 |
| 3 M Sodium acetate | Invitrogen | AM9740 |
| RNase inhibitor | New England Biolabs | M0314L |
| Linear acrylamide | Invitrogen | AM9520 |
| Isopropanol | Fisher Scientific | BP2618-212 |
| Chloroform | Alfa Aesar | J67241 |
| Proteinase K Solution | Invitrogen | 25530049 |
| 0.5 M EDTA | Invitrogen | AM9261 |
| 10% SDS | Sigma-Aldrich | 71736-100 ML |
| 10% Triton X-100 | Sigma-Aldrich | 93443-100 ML |
| 10XPBS | Gibco | 70013032 |

| Reagent/resource | Reference or source | Identifier or catalog number |
|---|---|---|
| Tris-HCl, pH 8.0 | Invitrogen | AM9855G |
| Nuclease-free water | Invitrogen | 10977-015 |
| Ethanol | Koptec | Cat. No. V1001 |
| Low-range ssRNA ladder | New England Biolabs | N0364S |
| 14–30 ssRNA Ladder Marker | Takara | 3416 |
| 10 mM ATP solution | New England Biolabs | P0756S |
| HEPES (pH 8.0) | Gibco | 15630080 |
| Ferrous ammonium sulfate | Sigma-Aldrich | 215406 |
| α-ketoglutaric acid | Sigma-Aldrich | K1128-25G |
| Bovine serum albumin (BSA) | Sigma-Aldrich | A7906-500G |
| Knock-out DMEM | Gibco | 10829 |
| Fetal bovine serum (FBS) | Gibco | 10437; Lot-2190737RP |
| GlutaMAX Supplement | Gibco | 35050061 |
| 100 X Streptomycin solution | Gibco | 15140 |
| Non-essential amino acids | Gibco | 11140 |
| 2-Mercaptoethanol | Gibco | 21985 |
| ESGRO® Recombinant Mouse LIF Protein | Sigma-Aldrich | ESG1106 |
| **Software** | | |
| SPORTS1.1 | https://github.com/junchaoshi/sports1.1 | |
| Kallisto tool | https://github.com/pachterlab/kallisto | |
| edgeR | https://bioconductor.org/packages/release/bioc/html/edgeR.html | |
| DAVID tool | https://davidbioinformatics.nih.gov/ | |
| Illustrator software | Adobe Inc. | |
| R | https://www.r-project.org/ | |
| **Other** | | |
| Illumina NovaSeq X plus | Illumina | |

## Animals

Animal experiments were conducted under the protocol and approval of the institutional animal care and use committees of the University of California, Riverside. Mice were given access to food and water ad libitum and were maintained on a 12 h light/12 h dark artificial lighting cycle. Mice were housed in cages at a temperature of 22–25 °C, with 40–60% humidity.

## Mouse sperm samples during aging

Mature sperm were collected from the cauda epididymis of male C57BL/6J mice, aged at 10, 30, 50, 70, and 90 weeks, and then were released into 5 ml of phosphate-buffered saline (PBS). This mixture was then incubated at 37 °C for 15 min, followed by filtration through a 40-μm cell strainer to remove tissue debris. To eliminate somatic cell contamination, the filtered sperm were incubated with a somatic cell lysis buffer (comprising 0.1% sodium dodecyl sulfate (SDS) and 0.5%

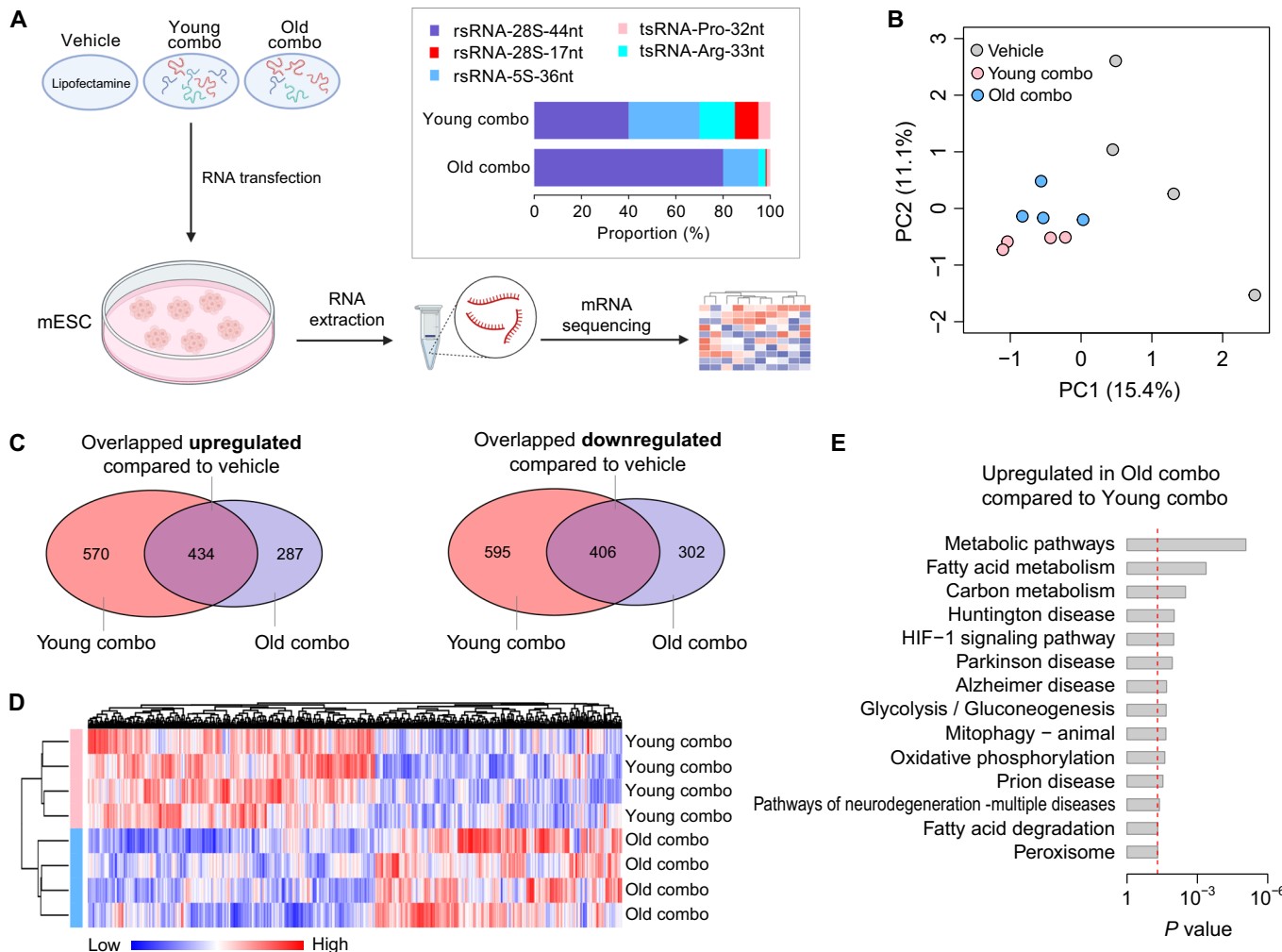

**Figure 4. Functional impact of age-related sperm sncRNAs on transcriptomic profiles in mouse embryonic stem cells.**

(A) Schematic of the experimental design: Synthetic RNA cocktails mimicking "young" or "old" sperm sncRNA profiles (composed of selected tsRNAs and rsRNAs with significant age-related differences in expression and length) were transfected into mESCs via lipofectamine, followed by RNA extraction 24 h post transfection, and mRNA sequencing analysis. Bar plots show the proportional composition (%) of key sncRNAs in the young combo (28S-rsRNA-44nt (40%); 28S-rsRNA-17nt (10%); 5S-rsRNA-36nt (30%); tsRNA-Arg-33nt (15%); tsRNA-Pro-32nt (5%)) and old combo (28S-rsRNA-44nt (80%); 28S-rsRNA-17nt (0.5%); 5S-rsRNA-36nt (15%); tsRNA-Arg-33nt (3%); tsRNA-Pro-32nt (1.5%)). (B) Principal component analysis (PCA) of the transcriptomic profile from mESCs transfected with vehicle, young combo, or old combo RNAs ($n = 4$ per group). (C) Venn diagrams illustrating overlap in upregulated (left) and downregulated (right) genes in young combo and old combo groups compared to the vehicle. (D) Heatmap of differentially expressed genes between young combo and old combo groups, showing clustering of replicates ($n = 4$ per group). (E) The top KEGG pathways ($P < 0.05$) associated with the upregulated genes in the old combo group relative to the young combo group. The red dashed line indicates the significance level of $a = 0.05$. (Modified Fisher's exact test). Source data are available online for this figure.

Triton X-100 in nuclease-free water) for 40 min on ice. Subsequently, the sperm were pelleted by centrifugation at $600 \times g$ for 5 min. The sperm pellet was then resuspended in 10 ml PBS, washed, and centrifuged twice at $600 \times g$ for 5 min each time. Finally, RNA isolation was performed on the precipitated sperm. Sperm age groups and sample information for sequencing are included in Table EV9.

## Human sperm samples

Study participants gave informed consent for semen samples to be used for research under University of Utah IRB # 0012049. In longitudinal cohort (cohort-1) with known fertility, eight donors each provided two semen samples across an interval of 6–23 years,

with the ages at which they provided these samples detailed in Fig. 3A; Table EV10. Smoking and alcohol use were exclusion criteria for donors in cohort-1. In the cross-sectional cohort (cohort-2), 47 individuals are included (25–51 years) detailed in Fig. 3B and Table EV11. Two participants in cohort-2 self-reported smoking (one regularly smokes cigarettes, the other smokes 1–2 cigars/month), and 24 self-reported some alcohol consumption (1–2 times/week), with no heavy drinkers included. Smoking and alcohol use information of cohort-2 is included in Table EV11. All semen samples were collected in our Andrology Laboratory (University of Utah School of Medicine) under the following quality criteria: (i) sperm concentration $\geq 15 \times 10^6$/ml; (ii) progressive motility $\geq 32\%$ (iii) normal morphology $\geq 14\%$. All semen samples were collected

by masturbation following 2–5 days of sexual abstinence and were subsequently cryopreserved in commercially available sperm cryopreservation media (TYB media; Irvine Scientific) and stored in liquid nitrogen until use in the study. The sizes of human cohorts are chosen based on the resource available at our disposal at the time of study.

## Sperm head isolation

Sperm head isolation was according to established protocols (Shi et al, 2025). For mouse sperm head, mature sperm were released from the cauda epidymis of male mice into 5 mL PBS and incubated at 37 °C for 15 min. For human sperm head, 1 mL frozen semen sample were thaw at 37 °C for 2 min then added to 4 mL PBS followed by incubation at 37 °C for 15 min. The suspension was filtered through a 40-µm cell strainer to remove tissue debris and centrifuged at 3000×*g* for 5 min. Pelleted sperm were incubated in lysis buffer (10 mM Tris-HCl, pH 8.0; 10 mM EDTA; 50 mM NaCl; 2% SDS; and 75 µg/mL proteinase K) for 15 min at room temperature, then centrifuged at 3000×*g* for 5 min. The pellet was collected, resuspended, washed in 5 ml PBS and centrifuged at 600×*g* for 5 min, repeated twice. Sperm head purity was confirmed by light microscopy before RNA extraction. For RNA isolation, add at least 500 µL TRIzol to the pellet and pass the suspension repeatedly through a 27 G needle until fully lysed with no visible precipitate.

## Total RNA isolation

TRIzol reagent (1 ml; Invitrogen, Cat. No. 15596018) was added to microtubes containing sperm and sperm head samples, followed by uniform vortexing. The samples were then incubated at room temperature for 5 min. To each milliliter of the sample, 200 µl of chloroform (Alfa Aesar, Cat. No. J67241) was added, vortexed for 15 s, incubated at room temperature for 2 min, and then centrifuged at 12,000×*g* for 15 min at 4 °C. The aqueous phase was transferred to a new microtube and mixed with an equal volume of isopropanol (Fisher Scientific, Cat. No. BP2618-212). After mixing, the samples were incubated at room temperature for 10 min, followed by centrifugation at 12,000×*g* for 10 min at 4 °C. The supernatant was discarded, and the pellet was washed with 1 ml of 75% ethanol (Koptec, Cat. No. V1001), then centrifuged at 7500×*g* for 5 min at 4 °C. The supernatant was removed, and the pellet was air-dried for 5 min. Finally, the pellet was resuspended in nuclease-free water, quantified, and either stored at −80 °C or used for further analyses.

## Isolation of 15–50 nt RNA fraction from total RNAs

The RNA sample, mixed with an equal volume of 2× RNA loading dye (New England Biolabs; B0363S), was incubated at 75 °C for 5 min. The mixture was loaded into 15% (wt/vol) urea polyacrylamide gel (10 ml mixture containing 7 M urea (Invitrogen; AM9902), 3.75 ml Acrylamide/Bis 19:1, 40% (Ambion; AM9022), 1 ml 10× TBE (Invitrogen; AM9863), 1 g l⁻¹ ammonium persulfate (Sigma–Aldrich; A3678-25G) and 1 ml l⁻¹ TEMED (Thermo Fisher Scientific; BP150-100)). The gel was run in a 1× TBE running buffer at 200 V until the bromophenol blue reached the bottom of the gel. After staining with SYBR Gold solution (Invitrogen; S11494), the gel that contained small RNAs of 15–50 nucleotides was excised based on small RNA ladders (New England Biolabs (N0364S) and

Takara (3416)) and eluted in 0.3 M sodium acetate (Invitrogen; AM9740) and 100 U ml⁻¹ RNase inhibitor (New England Biolabs; M0314L) overnight at 4 °C. The sample was then centrifuged for 10 min at 12,000×*g* (4 °C). The aqueous phase was mixed with pure ethanol, 3 M sodium acetate and linear acrylamide (Invitrogen; AM9520) at a ratio of 3:9:0.3:0.01. Then, the sample was incubated at −20 °C for 2 h and centrifuged for 25 min at 12,000×*g* (4 °C). After removing the supernatant, the precipitation was resuspended in nuclease-free water, quantified and stored at −80 °C or used for further processing. The RNAs were then separated into two halves, one for PANDORA-seq, and the other for traditional sncRNA-seq.

## PANDORA-seq and traditional sncRNA-seq

### PANDORA-seq

RNA fragments ranging from 15 to 50 nucleotides (nt) were incubated in a 50 µL reaction mixture containing 5 µL 10× PNK buffer (New England Biolabs; B0201S), 1 mM ATP (New England Biolabs; P0756S), 10 U T4PNK (New England Biolabs; M0201L) at 37 °C for 20 min. Following this incubation, the reaction mixture was added to 500 µL of TRIzol reagent to perform the RNA isolation procedure. Then the purified RNA was incubated in 50 µL reaction mixture containing 50 mM HEPES (pH 8.0) (Gibco (15630080) and Alfa Aesar (J63578)), 75 µM ferrous ammonium sulfate (pH 5.0), 1 mM α-ketoglutaric acid (Sigma–Aldrich; K1128-25G), 2 mM sodium ascorbate, 50 mg l⁻¹ bovine serum albumin (Sigma–Aldrich; A7906-500G), 4 µg ml⁻¹ AlkB, 2,000 U ml⁻¹ RNase inhibitor at 37 °C for 30 min. Subsequently, this reaction mixture was also added to 500 µL of TRIzol reagent to perform the RNA isolation procedure, followed by small RNA library construction and deep sequencing.

### Traditional sncRNA-seq

15–50 nt RNA fraction was directly processed for small RNA library construction and deep sequencing.

## Small RNA library construction and deep sequencing

The adapters were sourced from the NEBNext Small RNA Library Prep Set for Illumina (New England Biolabs, Catalog No. E7330S) and were ligated in sequence. First, a 3′ adapter was added under the following reaction conditions: incubation at 70 °C for 2 min, followed by 16 °C for 18 h. Subsequently, a reverse transcription primer was introduced under these conditions: 75 °C for 5 min, 37 °C for 15 min, and then 15 °C for 15 min. Next, a 5′ adapter mix was added, with the reaction conditions set at 70 °C for 2 min and then 25 °C for 1 h. First-strand cDNA synthesis proceeded at 70 °C for 2 min and then at 50 °C for 1 h. PCR amplification was carried out to enrich the cDNA fragments, utilizing the PCR Primer Cocktail and PCR Master Mix under the following conditions: an initial denaturation at 94 °C for 30 s; followed by 14–23 cycles of denaturation at 94 °C for 15 s, annealing at 62 °C for 30 s, and extension at 70 °C for 15 s; with a final extension at 70 °C for 5 min and then a hold at 4 °C. The libraries were then amplified and sequenced using the PE100 RUN configuration on Illumina NovaSeq X Plus platform by the University of California, San Diego IGM Genomics Center. The summaries of sequencing datasets for mouse (Table EV9) and human (Tables EV10 and 11) samples are provided along with each sample's age information.

## Processing the PANDORA-seq and traditional sncRNA-seq data

We used the *SPORTS* tool (v1.1) (Shi et al, 2018) to annotate and summarize the PANDORA-seq data and traditional sncRNA-seq with one mismatch tolerance. The "summary" files generated by *SPORTS* were used to quantify sncRNA read counts and reads per million (RPM) to measure the expression of the individual sncRNAs categories, including tsRNAs, rsRNAs, and miRNAs. For tsRNAs, only the RNA species derived from mature tRNAs were included for further analysis.

## RNA-seq transcriptomic data analysis

The *kallisto* tool (Bray et al, 2016) was applied to quantify genome-wide gene expression from the RNA-seq data. The read count matrix was further analyzed using the *edgeR* tool (Robinson et al, 2010) to identify the differentially expressed genes. In brief, the *TMM* method was employed for data normalization. The likelihood ratio test was used to prioritize the differentially expressed genes. The genes with $P < 0.05$ were deemed differentially expressed. The *DAVID* tool (Sherman et al, 2022) was used to identify the KEGG pathways associated with the differentially expressed genes.

## Small RNA transfection into mouse embryonic stem cells (mESCs)

Before transfection, $5 \times 10^6$ V6.5 mESCs were seeded into each well of 0.2% gelatin-coated six-well plate and incubated overnight (~16 h) with mESC media consisting of KO-DMEM (Gibco; 10829) supplemented with 15% FBS (Gibco; 10437; Lot-2190737RP), 2 mM GlutaMAX (Gibco; 35050), 100 U ml$^{-1}$ penicillin (Gibco; 15140), 100 µg ml$^{-1}$ streptomycin (Gibco; 15140), 100 uM non-essential amino acids (Gibco; 11140), 55 µM 2-mercaptoethanol (Gibco; 21985) and 1000 U ml$^{-1}$ LIF (Sigma-Aldrich; ESG1106). The transfection complex was prepared as follows: 4 µL of small RNA cocktail (100 µM) with 8 µL Lipofectamine™ Stem Reagent and 188 µL Opti-MEM was mixed by vortexing and incubated at room temperature for 15 min. The media was discarded, and 1800-µL fresh mESC media (excluding antibiotics) was added to the wells. About 200 µL lipofectamine–RNA transfection complex was then added to each well and incubated for 24 h at 37 °C under 5% $CO_2$. The final small RNA transfection concentration was 200 nM. For each transfection, four independent replicates were performed. Vehicle-only transfections were used as a control.

The small RNA cocktail groups are as follows:

Young-comb: 28S-rsRNA-44nt (80 nM); 28S-rsRNA-17nt (20 nM); 5S-rsRNA-36nt (60 nM); tsRNA-Arg-CCT (30 nM); tsRNA-Pro-CGG (10 nM)

Old-comb: 28S-rsRNA-44nt (160 nM); 28S-rsRNA-17nt (1 nM); 5S-rsRNA-36nt (30 nM); tsRNA-Arg-CCT (6 nM); tsRNA-Pro-CGG (3 nM)

The synthetic small RNA sequences are as follows:

tsRNA-Pro-CGG:
5'/Phos/rGrGrCrUrCrGrUrUrGrGrUrCrUrArGrGrGrGrUrAr-UrGrArUrUrCrUrCrGrCrUrU-2'3 c-Phos

tsRNA-Arg-CCT:
5'/Phos/rArGrGrGrArUrUrGrUrGrGrGrUrUrCrGrAr-GrUrCrCrCrArUrCrUrGrGrGrGrUrGrC-2'3 c-Phos

28S-rsRNA 44-nt:
5'/Phos/rCrUrCrGrCrUrGrCrGrArUrCrUrArUrUrGrArArAr-GrUrCrArGrCrCrCrUrCrGrArCrArCrArArGrGrGrUrUrUrG-2'3 c-Phos

28S-rsRNA 17-nt:
5'/Phos/rCrUrCrGrArCrArCrArArGrGrGrUrUrUrG-2'3 c-Phos

5S-rsRNA:
5'/Phos/rUrGrGrArArArCrCrGrCrCrUrGrGrGrArArUr-ArCrCrGrGrGrUrGrCrUrGrUrArGrGrCrU-2'3 c-Phos

At 24 h post-transfection, total RNA from mESCs was extracted with TRIzol (per manufacturer's instructions) and subjected to transcriptome-wide RNA-seq using a polyA enrichment strategy (Novogene), followed by bioinformatic analyses for differential gene expression and pathway discovery, as described previously (Shi et al, 2021).

## Statistical analyses

All the statistical analyses were conducted using the R programming platform. The *Spearman*'s rank correlation test, *Wilcoxon* test, and *Fisher*'s exact test were performed using the "cor.test", "wilcox.test", and "fisher.test" functions, respectively, with two-sided $P$ values being computed. The $F$-statistic was computed by the "aov" function. The principal coordinate analysis (PCoA) was performed using the "pcoa" function within the "ape" package. The dissimilarity indices were computed using the "vegdist" function within the "vegan" package based on the "clark" method. The principal component analysis (PCA) was performed using the "dudi.pca" function within the "ade4" package. The co-expression and gene expression heatmaps were generated by the "heatmap.2" function within the "gplots" package. The hierarchical clustering was performed using the "complete" method with "euclidean" distance. The Venn diagrams were plotted using the "venn.diagram" function within the "vennDiagram" package.

# Data availability

The sncRNA annotation pipeline *SPORTS* is available from GitHub (https://github.com/junchaoshi/sports1.1). RNA-seq datasets have been deposited in the Gene Expression Omnibus under the accession code GSE256182.

The source data of this paper are collected in the following database record: biostudies:S-SCDT-10_1038-S44318-025-00687-8.

# Peer review information

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

## Acknowledgements

This work is in part supported by NIH (R01HD092431 to QC, TZ, MP, and DM; R01ES032024 to QC and TZ; R35ES035015 to CZ, QC, and TZ; R01HD106112 to KI, JH, and AQ; R00HD111686 to XY), research funds from Induction Bio. (to QC and TZ), University of Utah startup funds (to QC) and Center for Genomic Medicine Pilot Award from University of Utah (to QC). This work includes data generated at the University of California, San Diego IGM Genomics Center funded by the NIH (P30DK063491, P30CA023100, and P30DK120515).

## Author contributions

**Junchao Shi**: Conceptualization; Formal analysis; Investigation; Methodology; Writing—original draft; Writing—review and editing. **Xudong Zhang**: Conceptualization; Data curation; Investigation; Writing—original draft; Writing—review and editing. **Chen Cai**: Data curation; Investigation; Writing—review and editing. **Shichao Liu**: Investigation. **Jiancheng Yu**: Data curation; Writing—review and editing. **Emma R James**: Resources; Investigation. **Lihua Liu**: Resources. **Benjamin R Emery**: Resources. **Megan R McMurray Bires**: Formal analysis. **Elizabeth Torres-Arce**: Investigation. **Hukam C Rawal**: Methodology; Writing—review and editing. **Joemy Ramsay**: Data curation. **Jason Kunisaki**: Data curation. **Changcheng Zhou**: Funding acquisition; Writing—review and editing. **David S Milstone**: Funding acquisition; Writing—review and editing. **Mary Elizabeth Patti**: Funding acquisition; Writing—review and editing. **Xiaoxu Yang**: Formal analysis; Writing—review and editing. **Tim G Jenkins**: Formal analysis; Writing—review and editing. **Aaron Quinlan**: Formal analysis; Funding acquisition; Writing—review and editing. **Bradley R Cairns**: Formal analysis; Writing—review and editing. **Paul Schimmel**: Formal analysis; Writing—review and editing. **James M Hotaling**: Resources; Funding acquisition; Writing—review and editing. **Kenneth I Aston**: Resources; Supervision; Funding acquisition; Writing—review and editing. **Tong Zhou**: Conceptualization; Formal analysis; Supervision; Funding acquisition; Methodology; Writing—original draft; Writing—review and editing. **Qi Chen**: Conceptualization; Formal analysis; Supervision; Funding acquisition; Writing—original draft; Writing—review and editing.

Source data underlying figure panels in this paper may have individual authorship assigned. Where available, figure panel/source data authorship is listed in the following database record: biostudies:S-SCDT-10_1038-S44318-025-00687-8.

## Disclosure and competing interests statement

JH is a clinical advisor for Induction Bio. The remaining authors declare no competing interests.

# Expanded View Figures

**Figure EV1. miRNA "aging cliff".**                                                                                                                                            ▶

miRNA profiles detected by PANDORA-seq show an aging cliff in intact sperm (**A–D**) and sperm heads (**E–H**) during mouse aging but is less prominent than the tsRNA/rsRNA profiles. (**A**) Illustrative figure showing the intact sperm collection at five time points (10-, 30-, 50-, 70-, and 90-week) during mouse aging. (**B, C**) Principal coordinate analysis (PCoA) of mouse sperm miRNA profiles showing that (**B**) PANDORA-seq, but not (**C**) traditional sncRNA-seq, identified an "aging cliff" during the 50–70-week transition in mouse sperm. Axis 1: the first principal coordinate; Axis 2: the second principal coordinate. (**D**) The ratio of between-group variance to within-group variance (*F*-statistic) is significantly higher in tsRNA/rsRNA group than that of the miRNA group in PANDORA-seq data, supporting that tsRNA/rsRNA profile shows a better classification power than miRNA profile between the demarcated stages (early 10-/30-/50-week vs. later 70-/90-week). The *P* value was computed by Wilcoxon test. (**E–H**) similar analyses on purified sperm heads as those of (**A–D**). For boxplot in (**D, H**), the bold horizontal line indicates the median of the data. The lower and upper box boundaries of the box indicate the 25th percentile (Q1) and 75th percentile (Q3) of the data, respectively. Accordingly, the interquartile range (IQR) is Q3-Q1. The lower and upper whiskers indicates the most extreme data points that fall within $Q1-1.5 \times IQR$ and $Q3 + 1.5 \times IQR$, respectively. For (**D**): $n = 1005$ for miRNA and $n = 81$ for tsRNA/rsRNA; For (**H**): $n = 950$ for miRNA and $n = 81$ for tsRNA/rsRNA.

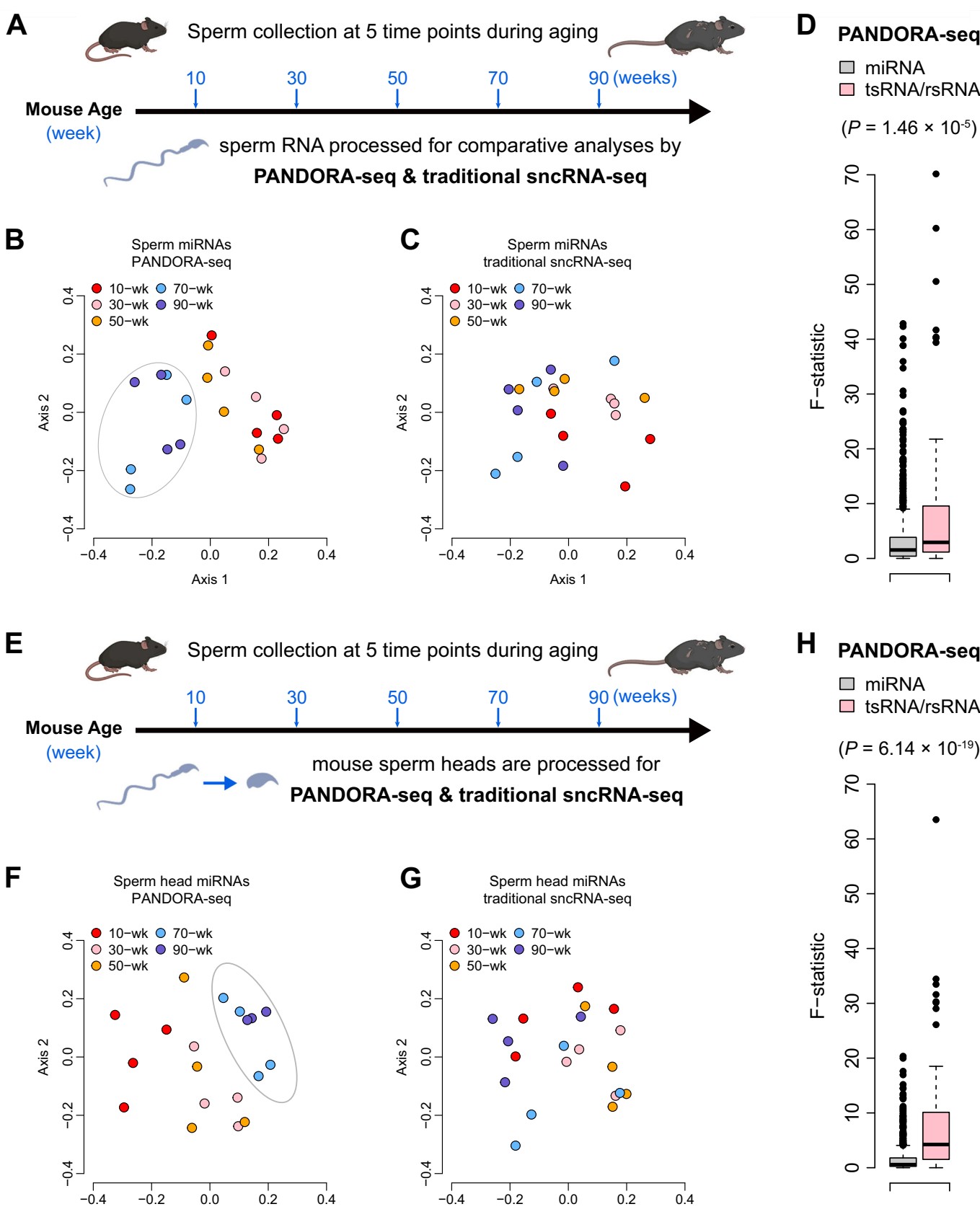

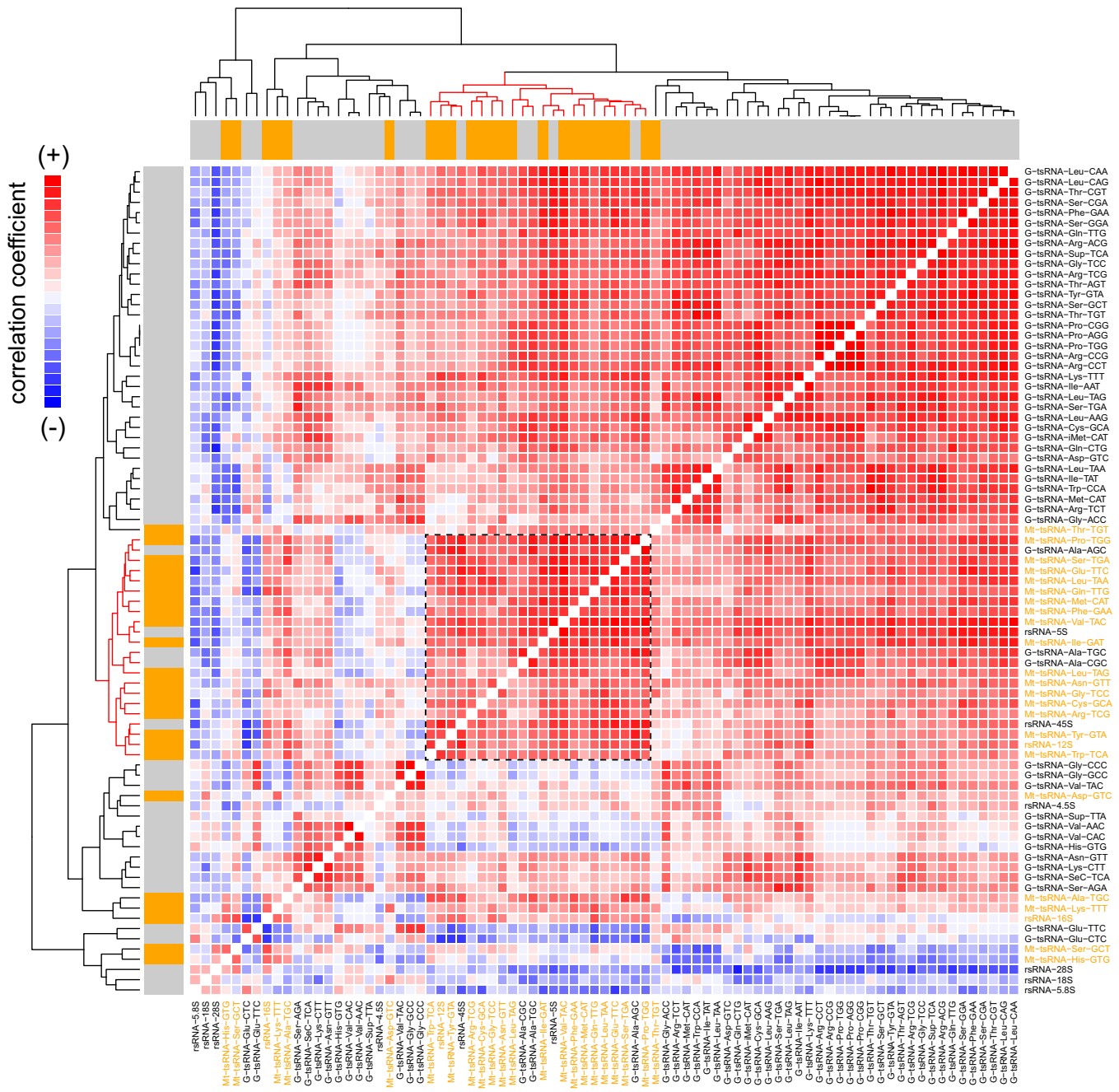

**Figure EV2.   Co-expression pattern of all the genomic tsRNAs/rsRNAs and mitochondrial tsRNAs/rsRNAs in the mouse de-membranated sperm heads.**

The colors in the heatmap represent the intensity of co-expression (i.e., Spearman's rank correlation coefficient) between the sncRNAs: red indicates positive co-expression, while blue indicates negative co-expression. This co-expression pattern is a magnified version of Fig. 1G, showing the identity of each tsRNA/rsRNA category on the heatmap. The enriched mt-tsRNA/rsRNA cluster (the dashed area in the middle) suggests that these mitochondrial sncRNAs are transcribed or regulated in a coordinated manner, which may reflect that, in the mature sperm, the genomic transcription is silent while the mitochondrial DNA transcription remains active. G-tsRNA genomic tsRNA, Mt-tsRNA mitochondrial tsRNA.

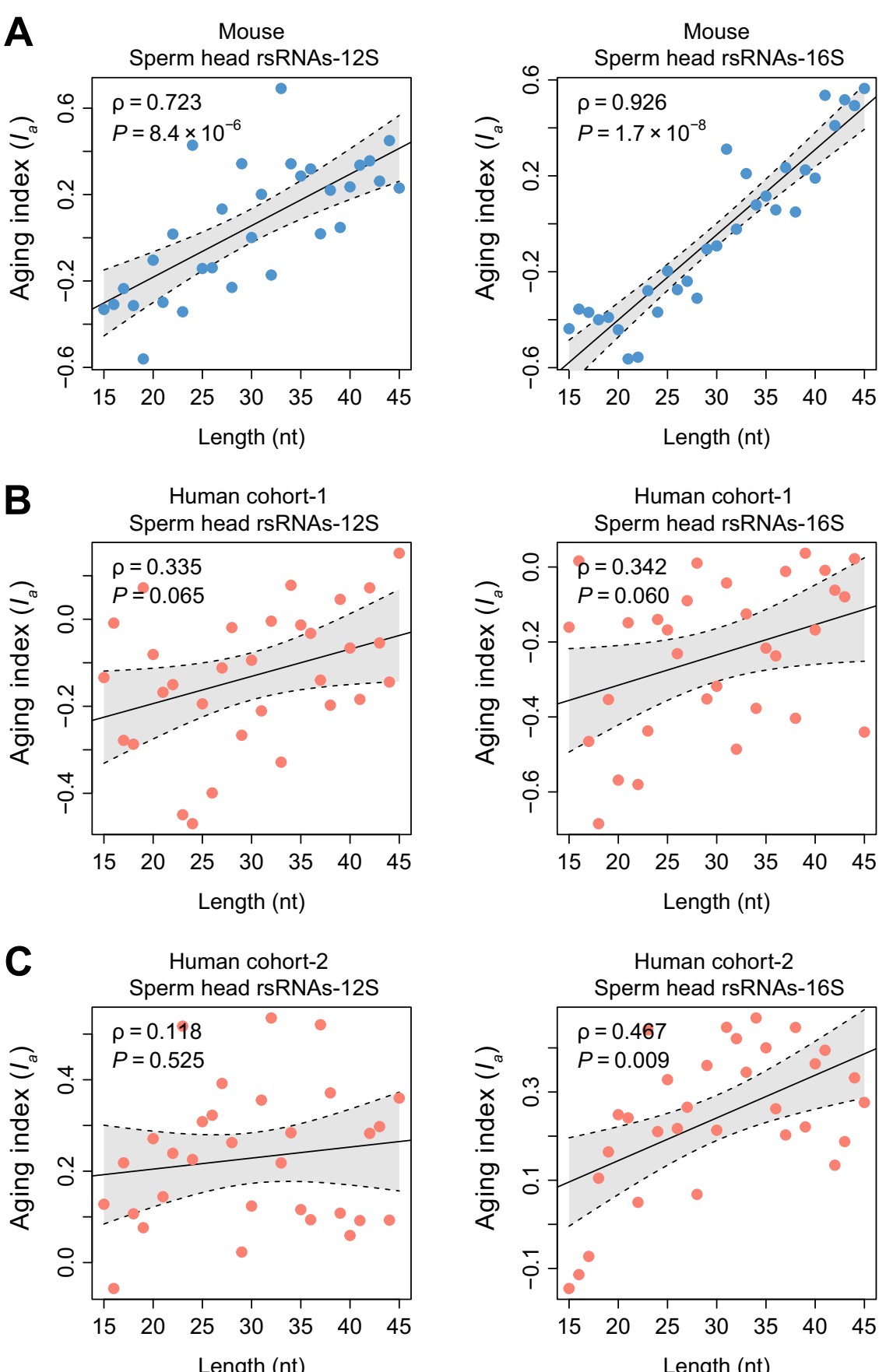

◀ **Figure EV3. Age-related length shift of mitochondrial rsRNAs (12S, 16S) in mouse and human sperm heads.**

RNA sample was extracted from de-membranated sperm heads from (**A**) mouse during 10–90 weeks as that of Fig. 2A, (**B**) human cohort-1, and (**C**) human cohort-2 as that of Fig. 3A, B, followed by PANDORA-seq. We calculated the association of expression (RPM) with age (Spearman's correlation), which we termed as aging index ($I_a$). Each dot represents the value of $I_a$ for the corresponding length. The scatter plots demonstrate the relationship between $I_a$ and RNA length, which was measured by Spearman's rank correlation coefficient ($\rho$) and the corresponding $P$ value. The solid lines depict linear regression fits.

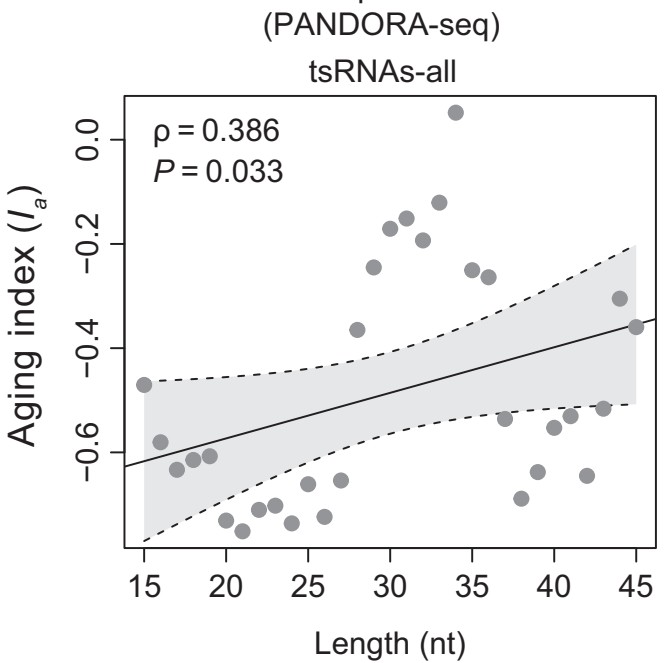

**Figure EV4.  Less prominent age-related length shift of tsRNAs in mouse sperm heads.**

Age-related length shift analyses of tsRNAs are similarly performed as that of rsRNAs in Fig. 2A, where sncRNAs were extracted from de-membranated sperm heads followed by PANDORA-seq. We calculated the association of expression (RPM) with age (Spearman's rank correlation), which we termed as aging index ($I_a$). Each dot represents the value of $I_a$ for the corresponding length. The scatter plots demonstrate the relationship between $I_a$ and RNA length, which was measured by Spearman's rank correlation coefficient ($\rho$) and the corresponding P value. The solid lines depict linear regression fits. The figure shows that the age-related length shift in tsRNAs is less prominent compared to that of rsRNAs in sperm heads.

