## [Peer Review File · The EMBO Journal]

Conserved shifts in sperm small non-coding RNA profiles during mouse and human aging

Junchao Shi, Xudong Zhang, Chen Cai, Shichao Liu, Jiancheng Yu, Emma James, Lihua Liu, Benjamin Emery, Megan McMurray, Elizabeth Torres-Arce, Hukam Rawal, Joemy Ramsay, Jason Kunisaki, Changcheng Zhou, David Milstone, Mary Elizabeth Patti, Xiaoxu Yang, Tim Jenkins, Aaron Quinlan, Bradley R. Cairns, Paul Schimmel, James Hotaling, Kenneth Aston, Tong Zhou, and Qi Chen

Corresponding authors: Qi Chen (qi.chen@hsc.utah.edu) , Tong Zhou (tongz@med.unr.edu), Kenneth Aston (ki.aston@hsc.utah.edu)

Review Timeline:

Submission Date:	26th Sep 25
Editorial Decision:	5th Nov 25
Revision Received:	4th Dec 25
Accepted:	12th Dec 25

Editor: Daniel Klimmeck

Transaction Report:

This manuscript was previously reviewed at another journal and transferred to The EMBO Journal. As EMBO Press has a transfer agreement (including the identities of the referees) with that journal, revision was invited based on the reports from that journal.

Point-to-point responses

Summary of Major Revisions

In response to the reviewers' constructive comments, we have substantially revised and expanded the manuscript. The main changes are as follows:

- **New Biological Insights:** According to editorial suggestion on journal priority/preference, we shifted the focus from a predictive biomarker (STAR signature) in the original submission to defining fundamental sncRNA pattern changes during sperm aging. This led to the discovery of a previously unrecognized rsRNA length shift in sperm heads (longer rsRNAs increase and shorter rsRNAs decrease with age) conserved in both mice and humans (new Figs. 2 & 3). We view this as a major advance in understanding mammalian sperm aging.
- **Expanded Human Validation:** We have increased from one longitudinal cohort (8 donors) to two independent cohorts (longitudinal: 8 donors; cross-sectional: 47 donors). Both human cohorts confirm the rsRNA length shift in sperm heads during aging (new Fig. 3) — consistent with mouse data, establishing this as a conserved feature of sperm aging.
- **New Functional Experiments:** We performed mESC transfection assays using RNA cocktails representing “young” and “old” sperm. The “old” cocktail induced transcriptomic changes in pathways related to metabolism, mitochondrial function, and neurodegeneration (new Fig. 4), mirroring phenotypes observed in offspring of aged fathers.
- **Enhanced Mitochondrial RNA Analysis:** We expanded the analyses of mitochondrial tsRNAs/rsRNAs, revealing synchronized pattern changes in sperm heads, and their high information density for aging discrimination despite their low abundance (new Fig. 1f-h).

Given the scope of these revisions, the manuscript has been expanded from a two-figure piece into a longer format with **four main figures**, more comprehensive data descriptions, and additional references.

We greatly appreciate the constructive suggestions from all reviewers, which have helped us substantially improved this study. Please find our detailed, point-to-point responses below.

Reviewer #1:

Overall comments:

SUMMARY OF KEY RESULTS

The manuscript reports interesting findings from longitudinal profilings of the sperm sncRNA payload in mice and humans. By using PANDORA-seq, the authors identify a sncRNA signature predictive of human and mouse age.

ORIGINALITY AND SIGNIFICANCE

Although the aging sperm epigenome has been already analysed (and all the relevant studies are also correctly mentioned in this ms), this is the first study to systematically look at the sperm sncRNA payload longitudinally in mice (across 5 different age-points). Also, the study incorporates data from human sperm samples donated at two different age-points from the same donors (for 8 donors and a total of 16 samples). The identified Sperm tsRNA and rsRNA (STAR) aging signature has biomarker potential for human aging and - as the authors state in the last sentence of the ms - holds the potential to guide informed reproductive decisions and reduce the transmission of age-related health issues to offspring. Unfortunately, the current study does not include any experimental validation of the physiological relevance of the STAR signature, neither for sperm biology, nor for the intergenerational consequences of paternal aging.

DATA AND METHODOLOGY

The data and the methodology are appropriate to the reported findings. More data would be instead needed to fully support the conclusions.

APPROPRIATE USE OF STATISTICS AND TREATMENT OF UNCERTAINTIES

The statistical methods are appropriate. The method to define the STAR signature could be more detailed and/or better explained.

CONCLUSIONS: ROBUSTNESS, VALIDITY, RELIABILITY

Most of the conclusions are supported by the data, and - as also shown in the past by the same corresponding author - the conclusions are robust, valid and reliable.

SUGGESTED IMPROVEMENTS: EXPERIMENTS, DATA FOR POSSIBLE REVISION

The following comments are provided with the sole goal of improving the already high quality of the study:

Overall Response: We greatly appreciate your overall positive comments on the novelty, significance, and quality of our study! With your constructive suggestions, we have now revised the manuscript substantially, incorporating new fundamental biological insights, expanded human cohorts, and functional evidence. Now the manuscript has been expanded from a two-figure *Brief communication* format to a more comprehensive four-figure *Letter format*. Please find our point-to-point responses below.

Specific comment 1:

As expected, the aging effect on the sperm sncRNA payload is visible by both PANDORA-seq and Traditional sncRNA-seq methods, although more pronounced (and better detectable) with PANDORA-seq as reported in Fig.1b-c (for both whole sperm and sperm heads). This might be due to the increased robustness and/or sensitivity of PANDORA-seq compared to traditional sncRNA-seq methods. It would be interesting to identify and indicate the corresponding samples in both sets (since both methods have been used for all the samples) and show a comparative analysis of the two methods (also by PCoA) to understand whether PANDORA-seq leads to a global or specific shift in the sperm sncRNA profile and to exclude potential external factors (operator, batch effect).

Response 1: Thanks for the great suggestion. In our experiments, individual RNA samples from mice (both intact sperm and sperm head) at each time point (10, 30, 50, 70, 90-week) were separated into two halves, one for traditional small RNA-seq, and the other for PANDORA-seq. The mouse RNA samples were processed by the same operator and sequenced in one batch.

As suggested, we have now labeled the sample numbers in PANDORA-seq vs. traditional sncRNA-seq settings and plotted them together in one PCoA (see figure below). The same numbers (1-4) as circles (PANDORA-seq) vs. triangles (traditional-seq) indicate pairs from the same mouse. The paired sample approach and single-batch sequencing minimize batch effects, as confirmed by the method-specific clustering rather than random variability. From the figure, we can see that PANDORA-seq and traditional-seq exhibit systematic differences on PCoA regarding overall tsRNA/rsRNA composition, likely a global shift due to PANDORA-seq's superior detection of modified sncRNAs. PANDORA-seq shows much better classification power in intact sperm and still better in sperm heads, where traditional-seq also shows some separation.

Specific comment 2:

Fig. 1d / 1e are interesting and show smallRNA (or biotype?) specific aging effect. For example, while the genomic tsRNA-Arg-CCC seems to be downregulated with age, the mt-tsRNA-His-GTG seems to be upregulated. The authors should extend this analysis to the entire biotype family to identify potential global shifts. What are the authors' takes on this phenomenon? And how do these results reconcile with recently published findings showing increased mitochondrial DNA transcription and mt-tRNA fragmentation in sperm upon acute dietary challenges and mitochondrial dysfunction?

Response 2: Thank you for the observations and especially the insightful questions on global pattern changes of mt-tsRNAs. In response, we performed expanded analyses of mt-tsRNAs and mt-rsRNAs, with key findings as follows (updated Fig. 1f-h; suppl. Fig. 2):

1) Although the sperm head samples are completely depleted of sperm tail (and thus mitochondria) (updated Fig. 1f, shown below), we consistently detected mt-tsRNAs and mt-rsRNAs. This suggests possible mitochondria-nucleus communication via small RNA transport.

2) In sperm heads, mt-tsRNAs and mt-rsRNAs were low in abundance (0.14% and 0.11%, respectively) yet highly correlated in their age-dependent changes (updated Fig. 1g, magnified in suppl. Fig. 2, both shown below). Such synchronization may in part, stem from mitochondrial transcription (circular, potentially active in mature sperm), resonating with reports that mt-tsRNAs/rsRNAs respond to dietary stress (*PLoS Biol* 2019, PMID: 31539371; *Antioxid Redox Signal* 2023, PMID: 36509450; *Nature* 2024, PMID: 38839949) and act as signal molecules regulating embryo development (*Nature* 2024, PMID: 38839949).

3) Importantly, despite their low levels, mt-tsRNAs and mt-rsRNAs in sperm heads robustly separated age groups in ordination analyses, mirroring genomic tsRNA/rsRNA aging patterns (updated Fig. 1h, shown below). This data suggest that the mitochondrial tsRNAs and rsRNAs contain high "information density"-even small quantities of these RNAs provide critical insights into sperm aging.

(f) Schematic illustrating de-membration of sperm and the image of purified sperm heads (depleted of tails and mitochondria), with pie chart showing the composition of sncRNAs in sperm heads: predominantly genomic rsRNAs (73.7%) and tsRNAs (26.0%), with minor mitochondrial tsRNAs (0.14%) and rsRNAs (0.11%).

(g) Correlation heatmap of all genomic tsRNA/rsRNA and mitochondrial tsRNA/rsRNA in the de-membrated sperm heads. The colors in the heatmap represent the intensity of co-expression (i.e., Spearman's rank correlation coefficient) between the sncRNAs: red indicates positive co-expression while blue indicates negative co-expression. Notably, the highly positively correlated clusters in the center square area (Fisher's exact test: P

$= 9.4 \times 10^{-8}$) are largely overlapped with mt-tsRNAs/rsRNAs, marked in orange color. The detailed identity of each tsRNAs and rsRNA category is shown in Suppl Fig. 2.

(h) Separated PCoAs for genomic tsRNAs, genomic rsRNAs, mitochondrial tsRNAs, and mitochondrial rsRNAs in sperm heads across the aging process. Mitochondrial tsRNAs and rsRNAs show demarcation power similar to genomic tsRNAs/rsRNAs, despite their relatively low expression level. The PCoA was performed based on the expression profile of the individual sncRNA species.

Supplementary Figure 2. Co-expression pattern of all the genomic tsRNAs/rsRNAs and mitochondrial tsRNAs/rsRNAs in the mouse de-membranated sperm heads. The colors in the heatmap represent the intensity of co-expression (i.e., Spearman's rank correlation coefficient) between the sncRNAs: red indicates positive co-expression, while blue indicates negative co-expression. This co-expression pattern is a magnified version of Fig. 1g, showing the identity of each tsRNA/rsRNA category on the heatmap. The enriched mt-tsRNA/rsRNA cluster (the dashed area in the middle) suggests that these mitochondrial sncRNAs are transcribed or regulated in a coordinated manner, which may reflect that, in the mature sperm, the genomic transcription is silent while the mitochondrial DNA transcription remains active. G-tsRNA: genomic tsRNA; Mt-tsRNA: mitochondrial tsRNA.

In addition to the clear pattern of mt-tsRNA/rsRNA in the sperm heads, we also examined the mt-tsRNA/rsRNA changes in the intact sperm. In contrast, mt-tsRNA/rsRNA patterns in intact sperm were more variable, with some species upregulated and others downregulated. We speculate that these mixed patterns may reflect two concurrent processes: (i) altered transcription of mt-tRNA/rRNA reflecting mitochondrial activity, and (ii) fragmentation of specific mt-tRNA/rRNAs driven by oxidative stress and RNase specificity, whereas the sperm mt-tsRNAs/rsRNAs serve as a functional readout of these combined forces. Environmental factors such as aging, dietary challenge, and obesity may differentially influence these processes, producing the observed patterns, with details warrant further investigations.

Specific comment 3:

PANDORA-seq has been recently developed by the corresponding author of this study to overcome the limitation of traditional sncRNA-seq methods with detecting highly modified sncRNAs in mature spermatozoa. Looking at the proportion of reads assigned to the different sncRNA biotypes (as reported in table S1) and assuming the sperm fraction does not carry contaminating somatic cells, PANDORA-seq seems to outcompete traditional

sncRNA-seq in detecting rsRNAs (av. 69.6% vs 32.3%) while being significantly less sensitive in detecting tsRNAs (av. 6.76% vs 23.76%). Does this mean that rsRNAs are more heavily modified than tsRNAs? And how do these modifications evolve with aging?

Furthermore, with both methods (for PANDORA-seq this is more pronounced in sperm heads) the fraction of reads assigned to tsRNAs declines and the rsRNA/tsRNA ratio increases with aging. This is an interesting result, also when extended to the potential role of sncRNAs in determining intergenerational epigenetic effects associated to increasing paternal age. How do the authors interpret these results? And is this maintained when looking at mt-sncRNAs (the fraction of reads assigned to mitochondrial sncRNAs is not reported in the table s1)? This last point would be particularly relevant in the light of the “high information density” exhibited by mt-sncRNAs in distinguishing sperm aging (as also stated by the authors).

Response 3: Thank you for the insightful observations and questions. Indeed, the differences in tsRNA and rsRNA proportions between PANDORA-seq and traditional sncRNA-seq reflect the impact of RNA modifications on detectability. Please find our detailed explanations below:

Why PANDORA-seq detects proportionally more rsRNAs than tsRNAs

Let us first reiterate how PANDORA-seq works:

PANDORA-seq removes two major classes of modifications that block small RNA detection:

(i) **Terminal modifications** (e.g., 3'-P, 2',3'-cP) that prevent adapter ligation.

(ii) **Internal base methylations** (e.g., m³C, m¹A, m¹G) that block reverse transcription.

Thus, a sncRNA carrying any one of these blocking modifications will be absent in traditional-seq but recovered by PANDORA-seq—an **all-or-none effect**, not a measure of “modification heaviness.”

That said, it does not necessarily mean rsRNAs are “more heavily modified” overall, they just need to carry **one** of those RNA modifications (either RT-blocking or ligation-blocking) to make them undetected in traditional-seq but uncovered in PANDORA-seq.

Regarding why PANDORA-seq detects proportionally more rsRNAs than tsRNAs, to our current knowledge:

(i) While tsRNAs harbor more diverse internal modifications, many do **not** block reverse transcription. Additionally, some tsRNAs are generated by enzymes (e.g., Dicer, RNase Z) that produce termini compatible with ligation in traditional-seq, making them detectable by traditional methods. Meanwhile other tsRNAs are generated by RNase T2 or RNase A family enzymes that produce termini that could not be ligated by traditional-seq but could be captured by PANDORA-seq.

(ii) rsRNAs are typically produced from rRNAs by RNase T2 or RNase A family enzymes, which often leave non-ligatable termini, but could be captured by PANDORA-seq. Consequently, a larger fraction of rsRNAs carry blocking modifications that prevent detection in traditional-seq, resulting in a more pronounced proportional increase under PANDORA-seq.

Thus, the higher rsRNA read increase in PANDORA-seq likely reflect their differences on biogenesis compared to tsRNAs.

Regarding the RNA Modifications during aging:

Our current data cannot directly address how RNA modifications change with aging because PANDORA-seq detects only the *presence or absence* of blocking modifications—it does not quantify specific modification types or total modification quantity. Therefore, read count changes cannot be directly translated into quantitative modification dynamics. Resolving this question will require dedicated, direct modification profiling in future studies.

tsRNA/rsRNA ratio changes and mitochondrial sncRNAs

We agree that the age-associated change of tsRNA/rsRNA ratio may have biological significance. To test this, we performed proof-of-principle experiments in which tsRNA/rsRNA cocktails representing “young” and “old” sperm were transfected into mESCs, followed by transcriptomic analysis. Our data show that these different combinations indeed have biological significance (see **response to your comment 5** and new Fig.4).

In addition, deeper analysis of mouse data revealed a what-we-believe-to-be a major discovery that sperm heads (but not the intact sperm) showing a clear rsRNA length shift during aging (longer rsRNAs increase and

shorter rsRNAs decrease with age). We have included these new insights in mice as new Fig. 2, and expanded to two human cohorts in new Figs. 3 (see **detailed response to comment 4.1**).

Regarding mitochondrial sncRNAs, we have now included the proportion of mitochondrial tsRNA/rsRNAs in an updated Suppl. Table 1 (see table below). We also performed deeper analyses of mt-tsRNAs/rsRNAs, yielding new insights that are presented in new Fig. 1f-h and Suppl. Fig. 2 (see response to comment 2 for details).

		Traditional sncRNA-seq					PANDORA-seq				
		miRNA (%)	GtsRNA (%)	MtsRNA (%)	GrsRNA (%)	MrsRNA (%)	miRNA (%)	GtsRNA (%)	MtsRNA (%)	GrsRNA (%)	MrsRNA (%)
Intact sperm	10-wk	6.322±1.253	27.539±1.547	0.532±0.108	29.253±1.438	1.201±0.230	0.097±0.007	6.839±0.665	0.060±0.013	73.783±0.823	0.167±0.023
	30-wk	6.440±1.076	31.172±1.800	0.451±0.041	27.015±1.746	1.140±0.127	0.076±0.005	8.994±0.809	0.052±0.004	70.412±1.036	0.135±0.009
	50-wk	7.309±1.689	24.297±0.530	0.401±0.037	30.753±1.835	1.154±0.204	0.112±0.008	6.981±0.684	0.034±0.004	70.201±1.166	0.127±0.014
	70-wk	7.212±1.472	14.851±1.608	0.602±0.114	36.328±8.434	1.232±0.178	0.162±0.016	4.680±0.300	0.093±0.006	67.350±2.111	0.257±0.021
	90-wk	9.566±2.107	18.160±2.497	0.761±0.188	32.125±6.966	1.222±0.259	0.104±0.003	5.997±0.739	0.116±0.012	65.536±0.728	0.301±0.036
Sperm head	10-wk	1.064±0.111	44.158±4.396	0.087±0.010	18.104±1.450	0.051±0.006	0.245±0.046	18.431±1.558	0.090±0.013	50.637±1.546	0.072±0.007
	30-wk	1.498±0.182	37.171±4.520	0.090±0.014	19.006±1.709	0.071±0.006	0.283±0.036	18.885±2.231	0.085±0.008	47.984±1.582	0.071±0.012
	50-wk	1.460±0.192	35.833±3.637	0.121±0.012	20.780±0.857	0.078±0.004	0.244±0.063	17.755±1.829	0.088±0.015	48.871±1.178	0.072±0.008
	70-wk	2.384±0.549	21.352±5.899	0.575±0.351	40.657±6.502	0.147±0.021	0.172±0.056	15.899±2.445	0.110±0.021	48.194±0.980	0.081±0.009
	90-wk	1.070±0.188	23.750±5.931	0.155±0.050	43.805±6.397	0.094±0.024	0.111±0.012	16.059±0.981	0.083±0.014	53.131±1.767	0.075±0.013

Specific comment 4 (4.1 & 4.2):

For the analysis of human sperm samples, the authors got access to an interesting - although small - cohort of men and two donations per person. Also, the authors decided to focus on sperm heads to analyse human samples. This is an important choice, as also claimed in the manuscript, due to the potential different physiological relevance of sncRNAs stored in sperm head vs cytoplasm. I do have two questions on this point: 4.1. Also based on their previous study (Shi et al. Nat.Cell Biol. 2021) showing distinct profiles in whole sperm vs sperm heads, why did the authors decide to study both sperm fractions for the murine samples in this study?

Response 4.1: Thanks for the positive comment on our human cohort! In the revised manuscript, we have now added a new cross-sectional human cohort (47 individuals), which further strengthens the robustness of our conclusions (see new Fig.3).

Our decision to analyze only sperm heads in human samples was based on two considerations:

Practical reason: Many archived human semen samples in the biobank (some stored for decades) exhibited substantial sample-to-sample variability. Issues included increased viscosity, residual surface debris resistant to washing, and the presence of cytoplasmic droplets that encapsulate diverse sncRNAs and could confound sperm-specific signatures. To minimize these sources of variability and ensure uniform RNA quality across samples, we isolated and analyzed only de-membrated sperm heads for human studies. We have incorporated this in the manuscript/methods.

Scientific reason: In mouse samples, we performed PANDORA-seq for both intact sperm and sperm heads as exploratory. This leads to the revelation that the RNA aging signatures in sperm heads contain more nuanced information that sensitively reflecting aging process: Notably, a pronounced rsRNA length shift (increased long rsRNAs, decreased short rsRNAs with age) was observed only in sperm heads, not in intact sperm (see new Fig. 2a,b). Importantly, both human cohorts (analyzed only from sperm heads) showed the same rsRNA length shift trend, indicating this is a conserved, cross-species phenomenon. This shift may reflect age-related changes in RNA processing enzyme activity or abundance, which warrants future investigation.

4.2. Are cytoplasmic sncRNAs differently modified compared to sncRNAs stored in the sperm head? Or, in other words, is it known whether sncRNAs acquire/lose modifications while being transferred to the sperm head? Is this process affected by aging?

Response 4.2: Thank you for raising these interesting questions. Currently, we do not have direct data on RNA modification status in cytoplasmic vs. head-localized sncRNAs. As we noted in Response 3, PANDORA-seq removes specific RT- and ligation-blocking modifications to recover sequences, producing an all-or-none detection effect rather than quantitative modification measurements and thus has limited power in predicting the overall quantitative RNA modification status. Addressing whether

modifications differ between compartments or change during transfer to the sperm head, and whether this is affected by aging, will require dedicated, direct modification profiling in future studies, which we view as an important next step.

Specific comment 5:

The STAR signature identified by the authors has an interesting predictive power for human aging. Does it also have functional relevance, for example, in an in vitro setting? Would the microinjection of the 15 most relevant rsRNA/tsRNA (the ones common between whole sperm and sperm heads) into wild-type zygotes be sufficient to modify early embryo gene expression and physiology? This last point is particularly relevant considering that the authors claim that the identified STAR signature has biomarker value for human aging and holds the potential to guide informed reproductive decisions and reduce the transmission of age-related health issues to offspring.

Response 5:

Thank you for the question on the functional relevance of age-associated tsRNAs/rsRNAs. To address this, we performed mESC transfection assays using RNA cocktails designed to mimic the “young” and “old” sperm RNA profiles. These cocktails included tsRNAs and rsRNAs that were (i) highly expressed, (ii) most significantly altered with aging, and (iii) shared between sperm and sperm heads in mice (see new Fig. 4a for composition).

Principal component analysis revealed clear separation of gene expression profiles among mock-transfected, young-combo, and old-combo groups (new Fig. 4b). Both RNA-treated groups differed substantially from the vehicle, with ~60% overlap in altered genes (Venn diagram, new Fig. 4c), yet also showed distinct gene expression signatures from each other (heatmap, new Fig. 4d).

Gene Ontology analysis highlighted that the old-combo specifically upregulated gene pathways linked to metabolic processes (e.g., fatty acid metabolism, glycolysis/gluconeogenesis), mitochondrial function (oxidative phosphorylation, mitophagy), and neurodegenerative disease pathways (e.g., Parkinson’s, Alzheimer’s, Huntington’s) (Fig. 4e). These pathways parallel phenotypes reported in offspring from aged sperm or from zygotic injection of RNAs from aged sperm, including metabolic and neurological disorders.

Together, these results provide **proof-of-principle evidence** that combinations of tsRNAs/rsRNAs reflecting young versus old sperm can elicit distinct transcriptional responses in mESCs (as an embryo proxy), supporting their potential functional role in mediating age-related phenotypes. The mechanistic details of which await further investigations, and the RNA modification status of tsRNAs/rsRNA may further complicate the situation.

Figure 4: Functional impact of age-related sperm sncRNAs on transcriptomic profiles in mouse embryonic stem cells

(a) Schematic of the experimental design: Synthetic RNA cocktails mimicking 'young' or 'old' sperm sncRNA profiles (composed of selected tsRNAs and rsRNAs with significant age-related differences in expression and length) were transfected into mESCs via lipofectamine, followed by RNA extraction 24h post transfection, and mRNA sequencing analysis. Bar plots show the proportional composition (%) of key sncRNAs in the young combo (28s-rsRNA-44nt (40%); 28s-rsRNA-17nt (10%); 5s-rsRNA-36nt(30%); tsRNA-Arg-33nt (15nM); tsRNA-Pro-32nt (5%)) and old combo (28s-rsRNA-44nt (80%); 28s-rsRNA-17nt (0.5%); 5s-rsRNA-36nt (15%); tsRNA-Arg-33nt (1.5%); tsRNA-Pro-32nt (1.5%)).

(b) Principal component analysis (PCA) of the transcriptomic profile from mESCs transfected with vehicle, young combo, or old combo RNAs (n=4 per group).

(c) Venn diagrams illustrating overlap in upregulated (left) and downregulated (right) genes in young combo and old combo groups compared to vehicle.

(d) Heatmap of differentially expressed genes between young combo and old combo groups, showing clustering of replicates (n=4 per group)

(e) The top KEGG pathways ($P < 0.05$) associated with the upregulated genes in the old combo group relative to the young combo group. The red dash line indicates the significance level of $\alpha=0.05$.

Regarding your suggestion to microinject synthetic tsRNAs/rsRNAs into zygotes: while we fully agree that zygotic RNA delivery can be informative, our prior works and that of others demonstrate caveats, let us explain below: Our group has originally used the method of zygotic microinjection of sperm RNA fraction (30-40nt) from high-fat diet father, resulting in offspring metabolic phenotypes (Science 2016, Nat Cell Biol 2018); this method has been similarly used by others: injecting of sperm RNA fraction (30-40nt) from aged mice (90-week) inducing phenotype in offspring mice (Aging Cell 2021 PMID: 34448534). These data support that the alterations in sperm RNAs during aging are indeed functionally significant.

However, when comes to the injection of individual or a combination of synthetic tsRNAs/rsRNAs, our previous own attempts with *synthetic* tsRNA injections failed to recapitulate the phenotype as we use the endogenous RNA fraction (30-40nt). We attribute this to missing endogenous RNA modifications, which are critical for tsRNA/rsRNA function but not yet fully decoded. For this reason, we have avoided drawing in vivo conclusions from synthetic RNA injections until the “modification code” is better understood — a cautionary stance we have advocated in multiple reviews/perspectives we wrote.

On the other hand, we believe our current mESC tsRNA/rsRNA transfection experiments strike a balance, which demonstrate functional differences between “young” and “old” RNA signatures in a controlled system, while avoiding overinterpretation from synthetic RNA injections. This approach may provide **proof-of-principle** evidence for functional relevance, and leaves the door open for future in vivo validation once the modification landscape is more fully resolved, as we mentioned in the results.

Reviewer #2:

Overall comment:

This is an interesting manuscript that describes a new physiological phenomenon which the authors describe as the “aging cliff” – this is a phenomenon where sperm after a certain age exhibit markedly altered patterns of small RNA fragments. Although the authors do not demonstrate or show a way how these these altered small RNA fragments could lead to age-associated birth defects or altered sperm functions, they propose that this might be a marker of youthful versus old sperm. It is very possible that the increased association with birth defects may actually occur due to the aging cliff. It's not clear that the aging cliff is disease relevant. Nevertheless, the story is still very interesting as this is a relatively pronounced and unexpected alteration in sperm non-coding RNA composition. It may turn out to be biologically/physiologically meaningful or driver of altered sperm functions. The majority of the changes cure to occur in mice around 70 weeks, and nicely, the authors have some data from human sperm samples that seems to support that this is a physiological phenomenon that might be clinically relevant. There does seem to be some evidence of correlation between age and RNA composition based on the authors “aging score.”

Overall, I think the studies will executed. It is interesting that aging leads to a marked change in RNA composition. The only question is whether this phenomenological finding is in any way linked to altered function.

Overall Response: We greatly appreciate your overall positive comments! Now the revised manuscript has incorporated new biological insights regarding rsRNA length shift during aging (new Fig.2), with validation in two independent human cohorts (new Fig.3), and functional evidence of tsRNA/rsRNA cocktails representing “young” and “old” sperm (new Fig.4), as described in the **Summary of Major Revisions** above. Please find our point-to-point responses below.

Major comment:

1. Do other tissues show the aging score or is this a sperm-specific phenomenon? I think this is important to define this phenomenon as driving sperm defects.

Response: Thank you for this important question. To our knowledge, the baseline sncRNA signatures in sperm/sperm heads are quite unique compared to other tissue and cell types, as shown in our previous PANDORA-seq paper (*Nat Cell Biol* 2025 PMID: 33820973), likely reflecting tissue/cell-specific biogenesis.

To further directly assess whether the age-associated tsRNA/rsRNA changes we report are sperm-specific, we performed a pilot PANDORA-seq analysis on mouse brain and muscle collected at 10 and 90 weeks (two biological replicates per time point) and further examined four representative tsRNAs/rsRNAs that show the most pronounced and consistent age-related changes in sperm/sperm heads (and are used in the functional assays of new Fig. 4).

As shown below, the age-related trajectories in brain and muscle were not consistent with those observed in sperm/sperm heads — in several cases, changes were absent or in the opposite direction. This supports our prediction that age-associated sncRNA patterns are tissue-specific, shaped by each tissue’s unique enzymatic landscape and environment. This data may also urge the need to study other tissue/cell-specific aging process with dedicated efforts.

Figure (for review only). Tissue-specific changes in representative age-associated tsRNAs/rsRNAs in mice. Boxplots show RPM values for four tsRNAs/rsRNAs at different ages across sperm, sperm heads (10, 30, 50, 70, and 90 weeks, n = 4 biological replicates per time point), brain and muscle (10 and 90 weeks; n = 2 biological replicates per time point). Age-related patterns in brain and muscle differ in magnitude and/or direction from those in sperm/sperm heads, supporting a sperm-specific sncRNA aging signature.

Minor comment:

1. The authors discuss sperm aging and the risk of birth defects. I don't think that this is actually 'sperm' aging. I think this refers to aging of spermatogonium - these cells age and are not synonymous with sperm. The sperm are only around for a few weeks and to the best of my knowledge advanced paternal age does not change the lifespan of a sperm. I can't help but wonder if the wrong cell type was used in this study. The proper cell type is spermatogonia. The study is interesting nonetheless, and the various ncRNAs in sperm cells may reflect changes to the spermatogonium, but this issue should be discussed carefully.

Response: Thank you for raising this point. We agree that aging can exert substantial effects on spermatogonia, which may in turn contribute to the sncRNA changes observed in mature sperm. At the same time, the sperm sncRNA landscape can also be shaped post-testicularly, for example through epididymal maturation, during which somatic cells deliver RNAs to sperm via exosomes. We have added discussion to the revised manuscript to reflect these possibilities:

“The source of these sperm sncRNA changes remains an open question. They may arise from altered biogenesis and regulation during testicular germ cell development, epididymal sperm maturation, or from sncRNAs delivered by somatic cells reflecting systemic aging, or from a combination of these factors^{10, 37-39}.”

From a translational perspective, mature sperm represent the functional vehicle of male reproduction and the most accessible sample type in clinical and reproductive settings. Focusing on mature sperm thus provides direct relevance for potential biomarker development and future diagnostic or therapeutic applications, while also capturing the integrated output of upstream processes, including any aging-associated changes in spermatogonia.

Reviewer #3:

Overall comment:

Aging impacts male fertility and the health of their offspring. This study aims to identify reliable biomarkers, with a focus on small non-coding RNAs (sncRNAs), to inform reproductive decisions and mitigate the risk of disorders in offspring. The authors utilized a novel sequencing method, PANDORA-seq, to examine sncRNAs—particularly tsRNA, rsRNA, and miRNA—in sperm samples from mice at five different ages (10, 30, 50, 70, and 90 weeks) and from eight human donors at two different ages. In the mice samples, they discovered an 'aging cliff,' characterized by significant changes in sncRNAs derived from tsRNA and rsRNA during the transition from 50 to 70 weeks. From the data, the authors suggest an aging sperm signature termed “STAR”, which effectively predicted aging in human sperm, highlighting the potential of PANDORA-seq for future reproductive technologies. Their work, presented as a Brief Communication, holds promise to stimulate further research and capture a broad audience interest. However, there are critical issues that should be addressed.

Overall Response: We appreciate your overall positive comments! Now the revised manuscript has incorporated new biological insights regarding rsRNA length shift during aging (new Fig.2), with validation in expanded two independent human cohorts (new Fig.3), and functional evidence of tsRNA/rsRNA cocktails representing “young” and “old” sperm (new Fig.4), as described in the **Summary of Major Revisions** above. Please find our point-to-point responses below.

Major points:

Comment 1:

1) The authors found drastic differences in expression of sncRNAs between 50 and 70 weeks in mouse sperm samples and insist that this phenomenon had not been detected with traditional sncRNA sequencing methods. However, there already is a report describing big differences in miRNA profiles between 12-month (i.e., 48-week) and 20-month (80-week) old mouse sperm samples of C57BL/6 strain by microarray analyses using commercial chips (Miyahara et al., *Sci Rep*, 2023). It is, of course, scientifically important that the authors reproduced the consistent fact, but they should respect originality in the field.

Response 1: Thank you for this comment. We have now cited relevant miRNA-related studies, including Miyahara et al., *Sci Rep*, 2023, in the introduction as part of the background, to acknowledge prior observations and place our work in the context of existing literature.

Comment 2:

The authors describe a purification protocol to 'de-membrane' human sperm by removing the tail and mention it

in the methods section. However, in the methods, the standard total RNA protocol is described, but no mention is made of a protocol that can 'de-membrane' sperm.

Response 2: Thank you for pointing this out. Our sperm de-membration protocols have been described and validated in our previous publications (*Cell Res* 2012 PMID: 23044802; *Nat Cell Biol* 2021 PMID: 33820973), and more recently in a methodological paper (*Nat Protoc* 2025 PMID: 40181099). We have now included a clear description of the de-membrating steps in the Methods section of the revised manuscript, with references to these protocols.

Comment 3:

The authors state that 30% of the STAR signature (12 out of 40 sncRNAs) consist of mitochondrial tsRNAs and that mitochondrial tsRNAs are always detected in de-membrated sperm heads, suggesting a shuttling mechanism between mitochondria and nucleus. Have the authors assessed the success ratio of the protocols that remove the tail and midpiece (including mitochondria) from the head, as depicted in Figure S3A? In sperm head samples, it is rather possible that the presence of mtsRNA is contamination from the midpiece (as there are two common mtsRNAs in mouse sperm head samples and whole sperm samples). If this is not verified, it cannot be said that the samples were not actually contaminated. Providing phase contrast images of demineralised (and possibly not demineralised) sperm head samples and % purity of sperm heads without midpiece and tail and with midpiece and tail would be sufficient evidence.

Response 3: Thanks for the suggestion. We can confirm that the purity of sperm heads using our protocols is very high with no visible tails and midpiece. As advised, we provide the phase contrast images (see below). Also, we have in the revised manuscript included the de-membrated sperm head image in the updated Fig. 1f.

Comment 4:

Was the quality of the sperm sample (including morphology etc.) assessed at the three time points? If this information is available, a table containing the requested parameters should be added. In addition, information on human subjects, such as smokers, non-smokers, diet and alcohol consumption, if available, should also be added. This is because these factors can induce oxidative stress in sperm, which in turn may alter the 'sperm RNA code' as described in the text.

Response 4: Thank you for this important suggestion.

All semen samples were collected in our Andrology Laboratory (University of Utah School of Medicine) under the following quality criteria:

- Sperm concentration $\geq 15 \times 10^6/\text{ml}$
- Progressive motility $\geq 32\%$
- Normal morphology $\geq 14\%$

For the longitudinal cohort in the initial submission, both smoking and alcohol use were exclusion criteria. For the additional cross-sectional cohort (47 individuals) included in the revised manuscript, only two participant self-reported smoking (one regularly smokes cigarette, the other smokes 1-2 cigar/month), and 24 self-reported some alcohol consumption (1-2 times/week), with no heavy drinkers included. Given these low frequencies and the lack of heavy use, the potential confounding effects of smoking or alcohol are expected to be minimal. We have

included the smoking and alcohol use information of cohort 2 in Suppl Table 11. Collection criteria for human donors has now been added to the Methods section of the revised manuscript.

Comment 5:

Are the scnRNA profiles observed in this study related to the development of health problems and disorders in the offspring, as described in the introduction? If it is only possible to determine relative sperm age, but not to predict the onset of health disorders in offspring, then the predictive potential of this technique is reduced.

Response 5: Thank you for raising this important point regarding the functional implications of the observed sncRNA profiles. In the revised manuscript, we performed mESC transfection assays using RNA cocktails representing “young” and “old” sperm (new Fig. 4). The “old” cocktail induced transcriptomic changes in pathways related to metabolism, mitochondrial function, and neurodegeneration—phenotypes that have been reported in offspring of aged fathers. These results provide proof-of-principle evidence that the age-associated sncRNA changes we observe are functionally relevant.

We agree that directly linking these sncRNA profiles to health outcomes in the offspring of our human donors would be critical for assessing predictive potential, but such information (especially offspring’s health data) is not currently available for our current cohorts. We hope this limitation would be kindly understood. As mentioned in our **Summary of Major Revisions**, we have shifted the focus from a predictive biomarker (STAR signature) in the original submission to defining fundamental sncRNA pattern changes during sperm aging, which may guide future in vivo studies assessing impact on offspring health.

Comment 6:

In regard with the discrepancy between mouse and human samples, it is very difficult to discuss any conclusion because the number of human samples are very small.

Response 6: Thank you for this important comment. In the revised manuscript, we have addressed the sample size limitation by expanding from one longitudinal cohort (8 donors) to two independent cohorts: a longitudinal cohort (8 donors) and a cross-sectional cohort (47 donors). Both cohorts consistently confirm the rsRNA length shift in sperm heads during aging (new Fig. 3), mirroring the fundamental sncRNA pattern change observed in mouse sperm—where longer rsRNAs increase and shorter rsRNAs decrease with age. The reproducibility across independent cohorts and species strengthens the conclusion that this length shift is a conserved feature of mammalian sperm aging.

Minor points:

1) Main text line 16: ‘Terimini’ should be changed to ‘Termini’ as Line 15.

Response: Thanks! We have corrected this.

Dear Dr Qi Chen,

Thank you again for the submission of your amended manuscript (EMBOJ-2025-122571-T) to The EMBO Journal. We have carefully assessed your manuscript, and the point-by-point response provided to the referee concerns that were raised during review at a different journal. In addition, and as mentioned before, we decided to send the revised version of your work back to the original reviewers for their reassessment with respect to technical robustness, conceptual advance and overall suitability of your work for publication in The EMBO Journal. We have received three re-reports which I enclose below. As you will see from the experts' comments, they are now in favour of the work and supportive of publication at The EMBO Journal, pending satisfactory revision.

We are thus pleased to inform you that we can offer to swiftly move forward towards acceptance of this work at The EMBO Journal, pending minor revision of the following remaining issues, which need to be adjusted in a re-submitted version.

Please consider the remaining points by the referee #1 carefully and introduce complementary experiments or adjust the data presentation and discussion of the results, where appropriate. i.p. Ref#1, pt.1 points to a relevant control experiment to consider.

We also need you to take care of a number of minor issues related to formatting and data annotation, which I will share shortly in a separate message, together with additional changes and requests by our production team for Source Data provision.

Please submit a revised version of the manuscript using the link enclosed below, addressing the advisor's comments.

As you might have seen on our web page, every paper at the EMBO Journal now includes a 'Synopsis', displayed on the html and freely accessible to all readers. The synopsis includes a 'model' figure as well as 2-5 one-short-sentence bullet points that summarize the article. I would appreciate if you could provide this figure and the bullet points.

Please let me know any time should you have additional questions regarding above points.

Thank you again for giving us the chance to consider your manuscript for The EMBO Journal, I look forward to hearing from you and receiving your final revised version of the manuscript.

Best regards,

Daniel Klimmeck

Daniel Klimmeck PhD
Senior Editor
The EMBO Journal.

>> Author information: resolve discrepancy for D. Milstone. (our online system) vs D. Milestone (manuscript text) and Mary Elizabeth Patti (our online system) vs Mary-Elizabeth Patti (manuscript text).

>> Please add up to five keywords to your study.

>> Please provide the main manuscript text as .docx file.

>> Author Contributions: Please remove the author contributions information from the manuscript text. Note that CRediT has replaced the traditional author contributions section as of now because it offers a systematic machine-readable author contributions format that allows for more effective research assessment. and use the free text boxes beneath each contributing

author's name to add specific details on the author's contribution.

More information is available in our guide to authors.

>> Adjust the title of the 'Competing Interests' section to 'Disclosure and Competing Interests Statement' and move after Acknowledgements.

>> Correct order of manuscript sections: Abstract / Keywords / Introduction / Results / Discussion / Methods / Data Availability / Acknowledgements / Disclosure and competing interests statement // References / Figure legends / Tables and their legends / Expanded View Figure legends

>> Please change the heading "Summary" to "Abstract".

>> Provide a completed Author Checklist.

>> Figure callouts: Please ensure that the figures and panels are called out in sequential order. Currently, Suppl. Fig 4 is called out before Suppl. Fig 3 and Suppl. Table is called out after Suppl. Tables 10 and 11.

>> Figures in separate files: Figures should be removed from the manuscript text and uploaded as individual, high resolution figure files. Please rename the suppl. figures Figure EV1 - EV4. The figure legends should stay in the manuscript text, after the References, and the EV figure legends should be under the heading "Expanded View Figure Files".

>> Please provide source data for the study as to the separate request e-mail by my office team.

>> Funding: please enter the funding following information in the list of funders in our online system: 'R01HD106112'; should research funds from 'Induction Bio, University of Utah startup funds and Center for Genomic Medicine Pilot Award from University of Utah' also be mentioned in the funders list in our system?

>> References: adjust reference format to EMBO Journal format, 10 authors et al, and place References after the Discussion, before figure legends.

>> Data availability section: Merge Code and Data availability sections. Provide public access to the codes and RNAseq datasets, including URL and dataset ID. Remove the privacy and related tokens.

>> Add a Reagents and Tools table to the Methods section, as a separate file using the existing template in the Guide For Authors, listing key reagents, experimental models, software and relevant equipment.

>> Avoid textual redundancy in the introduction, results and discussion sections with your earlier 2017 study (PMID 28094257).

>> Consider additional changes and comments from our production team as indicated below:

DATA CHECK:

- DAS:

1. Please note that the specific URL for GSE256182 dataset is not provided in the data availability statement.
2. Please note that the specific URL for GSE256182 dataset is not provided in the data availability statement.

- Figure legends:

1. Please indicate the statistical test used for data analysis in the legends of figures 2B, 3B, 4E
2. Please note that the box plots need to be defined in terms of minima, maxima, centre, bounds of box and whiskers, and percentile in the legends of figures 2C, S1 D, H
3. Please note that information related to n is missing in the legends of figures 2C, S1 D, H

Further information is available in our Guide For Authors: <https://www.embopress.org/page/journal/14602075/>

authorguide

Referee #1:

In today's world of declining birthrates and an aging population, accurately assessing sperm quality is crucial for ensuring fertility and the health of the next generation. This manuscript applies PANDRA-seq to profile sperm small RNAs across mouse aging and reports an "aging cliff", with a head-specific rsRNA length shift mirrored in human sperm cohorts. The topic is timely and of broad interest for reproductive epigenetics and biomarker discovery. However, several issues should be addressed to meet EMBO J's standards in mechanisms, statistics, human data rigor, and reproducibility.

Major comments

1) Purity of "sperm head-only" preparation and mitochondrial carryover

The claim of mitochondria-free heads conflicts with persistent detection of mt-derived small RNAs. It would be better to (i) quantify fraction purity (mtDNA qPCR vs nDNA; TOMM20/COXIV WB; EM of fractions), (ii) perform RNase protection {plus minus} detergent to distinguish luminal vs surface RNAs, and (iii) replicate the key signal with an orthogonal head-isolation method.

2) "Aging cliff" requires formal change-point inference and batch control

PCoA separation is suggestive but not sufficient to claim a discrete transition. The authors should fit segmented regression/change-point models with bootstrap CIs; they should report per-sample QC (RIN, yield, mapping), include sequencing-run/batch covariates, and test robustness to distance metrics. Moreover, length-class shifts should be validated by an orthogonal assay (e.g., northern/cap-PCR) across the 50-70-week window.

3) Functional assays: supra-physiologic and unclear mechanism

The RNA cocktails are delivered at 200 nM with engineered stoichiometries; effects at this exposure may not reflect zygotic conditions. The authors should provide dose-response down to physiologic levels, extend time-courses, include a second pluripotent line, and add length-matched scrambled/non-targeting controls plus spike-ins. They should also consider discussing why transfection was chosen over zygote/sperm microinjection for physiological relevance.

4) Human cohorts: metadata, covariates, and internal consistency

The reviewer understands that limited availability of human samples. However, the authors should reconcile age ranges across text/figures and analyze human data with appropriate covariate (abstinence time, BMI, smoking/alcohol, semen parameters, storage time/lot). The authors should use paired analyses should be used where applicable and mixed-effects models otherwise. They also need to report effect sizes with 95% CIs and FDR-controlled p-values per length bin/class.

5) Reproducibility and computational transparency

The Introduction cites Miyahara et al. (Sci Rep, 2023) profiling mouse sperm miRNAs in aging. The authors should report whether their PANDORA-seq recapitulates those miRNAs (and rank order), providing quantitative concordance (e.g., overlap, Spearman/Pearson, AUROC for detection). For EMBO J, the analysis must be fully reproducible. Please deposit:

Executable pipeline: all scripts with exact parameters and random seeds; software/version list; and a containerized environment (Docker/Singularity) or lockfiles. A workflow (Snakemake/Nextflow/Makefile) documenting raw→processed provenance would be ideal.

Data products: per-sample count tables stratified by read length and biotype; isomiR handling; UMI policy (if used); adapter/ligation-bias corrections; and criteria for miRNA calling/thresholds.

Alignment policies: multi-mapping rules (including rDNA/mtRNA reads), mismatch/indel tolerances, and any filtering of repeats.

Figure generation: code/notebooks to rebuild all panels from processed data.

Please deposit raw and processed data to GEO/SRA (with checksums and a clear README) and code/environment to a permanent repository (GitHub + Zenodo DOI). Statements such as "available upon reasonable request" are insufficient for EMBO J.

Minor comments

1) Figure standards and units

The authors should ensure consistent age units (weeks/years), identical y-axis scales across comparable panels, and explicit n per group on each plot.

2) Power and sampling justification

They should provide a priori or post hoc power analyses for key comparisons; clarify how sample sizes were chosen and

whether replicates are biological vs technical.

3) Terminology and definitions

rsRNA/tsRNA subclass boundaries (length/biogenesis) should be defined and mechanistic terms ("transport," "loading") should be avoided unless supported by direct evidence.

4) Methods clarity

The authors should briefly restate PANDORA-seq preprocessing steps side-by-side with traditional sncRNA-seq to aid comparison; specify one-mismatch tolerance rationale and how multi-mapping to rDNA/tRNA repeats was handled.

Referee #2:

I would like to thank the authors for the effort they've made in addressing all my comments. I find the age-dependent rsRNAs length difference a novel and interesting feature of sperm RNA biology and I'm looking forward to seeing more follow-up stories on this aspect and its biological and intergenerational impact.

Referee #3:

The authors have addressed the relatively minor concerns that I raised. I think this manuscript is suitable for EMBO J.

We thank the referees for positive comments! Please see our response to remaining comments below:

Referee #1:

Comment 1: Purity of "sperm head-only" preparation and mitochondrial carryover

The claim of mitochondria-free heads conflicts with persistent detection of mt-derived small RNAs. It would be better to (i) quantify fraction purity (mtDNA qPCR vs nDNA; TOMM20/COXIV WB; EM of fractions), (ii) perform RNase protection {plus minus} detergent to distinguish luminal vs surface RNAs, and (iii) replicate the key signal with an orthogonal head-isolation method.

Response 1: We appreciate the reviewer's scrutiny regarding the purity of the sperm head fraction. We agree that rigorous exclusion of mitochondrial contamination is essential. To address the request for confirming the elimination of mitochondrial compartment, we would like to emphasize the unique compartmentalization of mature sperm. Unlike somatic cells, where mitochondria are distributed throughout the cytoplasm, sperm mitochondria are strictly confined to the mitochondrial sheath within the midpiece of the tail. Therefore, the physical separation of the sperm head from the tail, verified by microscopy (and in the best form by electronic microscopy (EM) as advised by the reviewer), is the most direct and definitive confirmation of mitochondrial removal.

To this end, we utilized the rigorous chemical demembration strategy established in the field by Yanagimachi & Yan (*Biol Reprod* 2008, PMID: 18256326). Their foundational work demonstrated that chemical lysis (using strong detergents) completely strips the sperm of membranes and organelles, yielding "nude" nuclei. as shown in the **figure below** (Fig.2 from *Biol Reprod* 2008, PMID: 18256326), which directly addresses the reviewer's request for EM evidence of this purity standard. Based on the foundational protocol, our lab has been using a sperm head-only protocol using somatic cell lysis buffer containing 0.1% SDS and 0.5% Triton X-100 to achieve this level of purity, as shown in our several previous publications (*Cell Res* 2012 PMID: 23044802; *Nat Cell Biol* 2021 PMID: 33820973; *Nat Protoc* 2025 PMID: 40181099), and we have provided a figure (Fig.1f) in the current manuscript to show the high purity of sperm head.

FIG. 2. Electron micrographs of the heads of the mouse spermatozoa without treatment (A, B) or after treatment with lysolecithin and pronase (C, D). Note the LL + pronase treatment completely removed the acrosome, plasma membrane, and nuclear envelope from the sperm heads. nu, Nucleus; a, acrosome; plm, plasma membrane; and ne, nuclear envelope.

Furthermore, the detection of mitochondrial-derived small RNAs (mt-tsRNAs and mt-rsRNAs) within these purified heads is consistent with our prior observations using PANDORA-seq (*Nat Cell Biol* 2021 PMID: 33820973). Rather than indicating contamination from the tail (which is demonstrably absent), these data reflect a genuine biological phenomenon where specific small RNAs are transported from mitochondria to the nucleus. We chose to highlight these mt-tsRNAs/rsRNAs in the current manuscript not because of their abundance (which is low), but because they exhibit a striking sensitivity to the aging process. We believe reporting this distinct, age-dependent mitochondrial sncRNA change is valuable for the field as it points toward potential mitochondria-nucleus signaling pathways involved in sperm aging.

Comment 2: "Aging cliff" requires formal change-point inference and batch control

PCoA separation is suggestive but not sufficient to claim a discrete transition. The authors should fit segmented regression/change-point models with bootstrap CIs; they should report per-sample QC (RIN, yield, mapping), include sequencing-run/batch covariates, and test robustness to distance metrics. Moreover, length-class shifts should be validated by an orthogonal assay (e.g., northern/cap-PCR) across the 50-70-week window.

Response 2: In the manuscript, we used "aging cliff" as a descriptive term to highlight the dramatic shift in sncRNA profiles at specific age range: the 50-70week change is more dramatic than any other time window with the same 20-week interval.

we have now performed the quantitative "change-point" inference requested by the reviewer to statistically define this transition. Because our samples were collected at discrete 20-week intervals rather than on a continuous timescale, we utilized a resampling-based F-statistic analysis based on Axis 1 of PCoA. This metric rigorously quantifies the ratio of *between-group variability* (the difference between the age groups before and after the "cliff") to *within-group variability* (the variance among biological replicates within each group). A higher F-statistic indicates a sharper, more distinct biological separation.

To test the robustness of the "cliff," we computed F-statistics for every adjacent age interval (10 vs 30, 30 vs 50, 50 vs 70, and 70 vs 90 weeks) using a resampling approach to generate confidence distributions.

As shown in the **Figure** below, the results clearly show that:

1. The F-statistic for the 50-70 week interval in PANDORA-seq data is substantially higher (shifting far to the right) compared to all other intervals. This statistically confirms that the molecular changes occurring between 50 and 70 weeks are distinct from the gradual drift observed at other ages.
2. The high value F-statistic at 50-70 weeks is clearly observed in PANDORA-seq (both whole sperm and sperm heads) but is much weaker or absent in traditional sncRNA-seq, further validating that PANDORA-seq captures modified RNA signatures that define this aging transition.

Figure: Resampling-based F-statistic analysis across aging intervals. The histograms show the distribution of F-statistics derived from 1,000 rounds of resampling test. In each round of resampling test, the samples were reshuffled and randomly assigned to an age group. Note the distinct, high-magnitude shift in the 50-70 week interval for PANDORA-seq samples (both intact sperm, and sperm heads), statistically confirming this window as the primary "aging cliff."

For the length shift validation of specific rsRNAs using other methods (e.g., northern blot) and to trace the origin of such regulation such as age-dependent enzymatic changes, we are currently keeping exploring these directions, but we believe these would be out of the scope of the current manuscript.

Comment 3: Functional assays: supra-physiologic and unclear mechanism

The RNA cocktails are delivered at 200 nM with engineered stoichiometries; effects at this exposure may not reflect zygotic conditions. The authors should provide dose-response down to physiologic levels, extend time-courses, include a second pluripotent line, and add length-matched scrambled/non-targeting controls plus spike-ins. They should also consider discussing why transfection was chosen over zygote/sperm microinjection for physiological relevance.

Response 3: Regarding the choice of transfection over zygote microinjection: We respectfully refer the reviewer to our detailed explanation in the previous revision response (specifically Response 5), where we have addressed why zygotic injection of synthetic RNAs is not a suitable strategy for this study. As stated previously, our own prior attempts (and those of others) with synthetic tsRNA injections failed to recapitulate phenotypes observed with endogenous RNA fractions. We attribute this failure to "missing endogenous RNA modifications, which are critical for tsRNA/rsRNA function but not yet fully decoded". Consequently, proceeding with zygotic injection of unmodified synthetic cocktails risks generating artifactual results. As we emphasized in our previous response, we have adopted a cautionary stance to avoid drawing in vivo conclusions from synthetic RNAs until the modification code is resolved.

Regarding dosage and additional cell lines: Consistent with our previous response, the mESC transfection assay was designed to provide "proof-of-principle evidence that combinations of tsRNAs/rsRNAs reflecting young versus old sperm can elicit distinct transcriptional responses". The concentration used (200 nM) is standard for ensuring efficient transfection in this cell type to detect these relative differences. Since this is a Resource article aimed at defining the "aging cliff" and providing a dataset for the community, we believe the current functional data, which shows specific upregulation of metabolic and neurodegenerative pathways mirroring offspring phenotypes, would be sufficient to demonstrate biological potential in an in vitro cell model (mESC) without requiring exhaustive in vivo modelling that is currently limited by the lack of proper RNA modifications on synthetic tsRNAs/rsRNAs.

Comment 4: Human cohorts: metadata, covariates, and internal consistency

The reviewer understands that limited availability of human samples. However, the authors should reconcile age ranges across text/figures and analyze human data with appropriate covariate (abstinence time, BMI, smoking/alcohol, semen parameters, storage time/lot). The authors should use paired analyses where applicable and mixed-effects models otherwise. They also need to report effect sizes with 95% CIs and FDR-controlled p-values per length bin/class.

Response 4: Thanks for the further questions. Regarding covariates and metadata, as detailed in our previous response (Response 4 to Reviewer 3), we have strictly defined our cohorts to minimize confounding factors. Specifically:

Cohort 1 (Longitudinal): Smoking and alcohol use were exclusion criteria.

Cohort 2 (Cross-sectional): Only two participants self-reported smoking and no heavy drinkers were included, as detailed in Supplementary Table 11.

Given these strict criteria and the sample size limitations inherent to human sperm RNA studies, stratifying by these minor covariates would be statistically underpowered given our current cohort size. However, the fact that the rsRNA length shift is observed with such high significance (Fig.3) in both cohort strongly suggests that the aging signal is the dominant driver, overriding other potential environmental variances.

We agree that paired analysis is valuable for longitudinal data, and we tested standard paired samples with statistic significant outcome, but we chose to present the Spearman correlation (Aging Index) as shown in Fig.3, because our goal in this Resource article is to establish a cross-population aging clock. A paired test could only answer the question of "does it change during aging", whereas the correlation analysis in Figure 3 provides much clearer data showing the linear correlation of rsRNA length shift during aging across the population. We believe this presentation provides the most intuitive and robust demonstration of the conserved rsRNA length shift as a universal biomarker.

Comment 5: Reproducibility and computational transparency

The Introduction cites Miyahara et al. (Sci Rep, 2023) profiling mouse sperm miRNAs in aging. The authors should report whether their PANDORA-seq recapitulates those miRNAs (and rank order), providing quantitative concordance (e.g., overlap, Spearman/Pearson, AUROC for detection).

Response 5: We appreciate the reviewer’s suggestion to benchmark our findings against existing literature. We have now performed a direct reanalysis of the miRNA changes reported by Miyahara et al. (Sci Rep, 2023) to assess concordance with our PANDORA-seq dataset.

First, it is important to note the platform differences: Miyahara et al. utilized miRNA microarrays at three time points (3, 12, and 20 months), whereas we employed PANDORA-seq at five time points (10, 30, 50, 70, and 90 weeks). Our time points align approximately as follows: (3 months ≈ 10 weeks; 12 months ≈ 50 weeks; 20 months ≈ 90 weeks)

As summarized in the table below, we found substantial overlap in the specific miRNAs altered during aging. Despite the platform differences (microarray vs. PANDORA-seq), many candidates identified by Miyahara et al. also show statistically significant dysregulation in our PANDORA-seq data at comparable age intervals.

Data from Miyahara, Sci Rep, 2023 (microarray)			Data from us (PANDORA-seq, mouse intact sperm)		
comparison	ID	p	comparison	ID	p.1
3M vs 12M	mmu-miR-877-5p	0.0084	10 vs 50 week	mmu-mir-877	0.03042139
3M vs 20M	mmu-miR-10a-5p	0.0011	10 vs 90 week	mmu-mir-10a	4.17E-05
3M vs 20M	mmu-miR-677-3p	0.0013	10 vs 90 week	mmu-mir-677	0.00781649
3M vs 20M	mmu-miR-195a-3p	0.0016	10 vs 90 week	mmu-mir-195a	0.02590748
3M vs 20M	mmu-miR-7054-5p	0.0074	10 vs 90 week	mmu-mir-7054	0.04458398
3M vs 20M	mmu-miR-690	0.0132	10 vs 90 week	mmu-mir-690	0.00110793
3M vs 20M	mmu-miR-7033-5p	0.031	10 vs 90 week	mmu-mir-7033	6.63E-06
3M vs 20M	mmu-miR-16-1-3p	0.0369	10 vs 90 week	mmu-mir-16-1	0.02565363
3M vs 20M	mmu-miR-3535	0.0431	10 vs 90 week	mmu-mir-3535	2.58E-05
12M vs 20M	mmu-miR-10a-5p	4.00E-04	50 vs 90 week	mmu-mir-10a	4.17E-05
12M vs 20M	mmu-miR-677-3p	0.0017	50 vs 90 week	mmu-mir-677	0.00781649
12M vs 20M	mmu-miR-10b-5p	0.0023	50 vs 90 week	mmu-mir-10b	0.00012247
12M vs 20M	mmu-miR-23a-3p	0.0026	50 vs 90 week	mmu-mir-23a	0.00136242
12M vs 20M	mmu-miR-195a-3p	0.0026	50 vs 90 week	mmu-mir-195a	0.02590748
12M vs 20M	mmu-miR-690	0.0039	50 vs 90 week	mmu-mir-690	0.00110793
12M vs 20M	mmu-miR-3535	0.0089	50 vs 90 week	mmu-mir-3535	2.58E-05
12M vs 20M	mmu-miR-30a-5p	0.0169	50 vs 90 week	mmu-mir-30a	3.94E-07
12M vs 20M	mmu-miR-29a-3p	0.0245	50 vs 90 week	mmu-mir-29a	0.00043813
12M vs 20M	mmu-miR-16-1-3p	0.025	50 vs 90 week	mmu-mir-16-1	0.02565363
12M vs 20M	mmu-miR-145a-5p	0.0342	50 vs 90 week	mmu-mir-145a	0.00304231

Most interestingly, our reanalysis of the Miyahara et al. data reveals a pattern that strongly supports our "aging cliff" hypothesis. In their study, the number of differentially expressed miRNAs is relatively small in the 3 vs. 12-month comparison (roughly 10-50 weeks) but increases dramatically in the 12 vs. 20-month comparisons (roughly 50-90 weeks). This suggests that in their dataset—just as in ours—the most profound transcriptomic remodeling occurs in the latter half of the lifespan (post-12 months/50 weeks). Thus, their microarray data provides independent, external validation of the "aging cliff" phenomenon we define here using PANDORA-seq.

Comment: For EMBO J, the analysis must be fully reproducible. Please deposit:

- Executable pipeline: all scripts with exact parameters and random seeds; software/version list; and a containerized environment (Docker/Singularity) or lockfiles. A workflow (Snakemake/Nextflow/Makefile) documenting raw→processed provenance would be ideal.
- Data products: per-sample count tables stratified by read length and biotype; isomiR handling; UMI policy (if used); adapter/ligation-bias corrections; and criteria for miRNA calling/thresholds.
- Alignment policies: multi-mapping rules (including rDNA/mtRNA reads), mismatch/indel tolerances, and any filtering of repeats.
- Figure generation: code/notebooks to rebuild all panels from processed data.

Please deposit raw and processed data to GEO/SRA (with checksums and a clear README) and code/environment to a permanent repository (GitHub + Zenodo DOI). Statements such as "available upon reasonable request" are insufficient for EMBO J.

Response: Thanks. We have followed the EMBO J guidelines from the editor and provide all materials/source files etc compliant to the journal policy.

Minor comments

1) Figure standards and units

The authors should ensure consistent age units (weeks/years), identical y-axis scales across comparable panels, and explicit n per group on each plot.

Response: Thanks, we have paid attention to make sure these details are correct

2) Power and sampling justification

They should provide a priori or post hoc power analyses for key comparisons; clarify how sample sizes were chosen and whether replicates are biological vs technical.

Response: The sizes of human cohorts are chosen based on the resource available at our disposal at the time of study. All the replicates shown in the manuscript are biological replicates.

3) Terminology and definitions

rsRNA/tsRNA subclass boundaries (length/biogenesis) should be defined and mechanistic terms ("transport," "loading") should be avoided unless supported by direct evidence.

Response: We have considered these suggestions and made the wording accurately reflect our intended meanings.

4) Methods clarity

The authors should briefly restate PANDORA-seq preprocessing steps side-by-side with traditional sncRNA-seq to aid comparison; specify one-mismatch tolerance rationale and how multi-mapping to rDNA/tRNA repeats was handled.

Response: We have added brief introduction of the core process of PANDORA-seq, and have referred to the details of the mapping strategy/protocol to our recently published papers (*Nat Cell Biol* 2021 PMID: 33820973; *Nat Protoc* 2025 PMID: 40181099)

Referee #2:

I would like to thank the authors for the effort they've made in addressing all my comments. I find the age-dependent rsRNAs length difference a novel and interesting feature of sperm RNA biology and I'm looking forward to seeing more follow-up stories on this aspect and its biological and intergenerational impact.

Response: We appreciate the reviewer's constructive comments and support for publication!

Referee #3:

The authors have addressed the relatively minor concerns that I raised. I think this manuscript is suitable for *EMBO J*.

Response: We appreciate the reviewer's constructive comments and support for publication!

We hope our revised manuscript has meet all the publication criteria for *EMBO J*, and we look forward to hearing from you!

Warmest wishes

Qi

--

Qi Chen, MD, PhD.
Associate Professor of Urology/Human Genetics
Molecular Medicine Program
University of Utah School of Medicine
Website: <http://qichen-lab.info/>

Dear Dr Chen,

Thank you for submitting the revised version of your manuscript. I have now evaluated your amended manuscript and concluded that the remaining minor concerns have been sufficiently addressed.

I am thus pleased to inform you that your manuscript has been accepted for publication in the EMBO Journal.

Kind regards,

Daniel Klimmeck

Daniel Klimmeck, PhD
Senior Editor
The EMBO Journal
EMBO
Postfach 1022-40
Meyerhofstrasse 1
D-69117 Heidelberg
contact@embojournal.org

Please note that it is The EMBO Journal policy for the transcript of the editorial process (containing referee reports and your response letters) to be published as an online supplement to each paper. If you should prefer removal of any referee-only figures included in the point-by-point response(s), e.g. because they may still be used for future publication or because they have been reproduced from published work by others, please do let us know immediately via response email.

More information is available here: <https://link.springer.com/partners/embo-press/editorial-policies#Peer%20review>